# PARTITION-LOSSES FINE-TUNING: CONTAMINATION-ROBUST BACKDOOR UNLEARNING

## ABSTRACT

Large-scale training data and third-party checkpoints make training convenient but also leave room for poisoning-based backdoor attacks. These attacks embed a backdoor through data poisoning in the training set: the infected model behaves normally on clean inputs but predicts an attacker-chosen label whenever the trigger appears. The stealthiness poses risks for security-sensitive deployment and model reuse. Post-training fine-tuning has become a practical default defense as it is computationally efficient and does not require control over the original training pipeline. However, existing fine-tuning methods rely on a clean set to unlearn the backdoor indirectly. This assumption is fragile in reality: curation errors or undetected triggers can contaminate the perfectly "clean" set. As a result, state-of-the-art clean-only fine-tuning often fails to purify the backdoor behavior while maintaining the original functionality. We propose Partition-Losses Fine-Tuning (PL), a simple, architecture- and domain-agnostic loss modification that leverages both mostly-clean and flagged mostly-malicious samples. PL jointly minimizes benign loss and maximizes target-class loss, explicitly pushing the model away from the implanted trigger-to-target association. Comprehensive experiments show that PL matches or surpasses clean-only fine-tuning methods under the same computational budget while halving the required clean samples. Crucially, PL remains effective under realistic contamination of both fine-tuning sets and is stable across hyperparameter choices and data availability.

## 1 INTRODUCTION

Deep neural networks (DNN) have achieved state-of-the-art performance across diverse tasks (Canziani et al., 2016). This success is powered by large-scale training data and third-party pre-training, which provide rich representations and accelerate progress. At the same time, this reliance creates security risks. Uncurated data can be poisoned, and pretrained checkpoints may already contain hidden vulnerabilities, which can persist unnoticed in downstream deployment. In particular, poisoning-based backdoor attacks inject a hidden trigger-to-target mapping through the training data. This enables models to behave normally on clean inputs while misclassifying triggered inputs into an attacker-chosen label (Carlini et al., 2024; Goldblum et al., 2022; Shejwalkar et al., 2022). Because the model's accuracy on clean data remains high, these attacks are difficult to detect with standard validation, posing significant challenges to security-sensitive deployment and model reuse (Carlini & Terzis, 2022; Qin et al., 2023).

Various defenses have been proposed targeting different stages of the pipeline. Pre-training defenses aim to filter suspected poisoned samples or neutralize poisoned samples, for example, through anomaly detection, trigger localization before training (Doan et al., 2020; Liu et al., 2017). In-training defenses intervene in optimization to inhibit backdoor injection (Wu et al., 2022). However, both approaches require access to and control over the training process, which is often impractical and costly for large models (Min et al., 2023; Huang et al., 2022; Zeng et al., 2021). Post-training defenses, particularly fine-tuning, have therefore become a practical default. It operates on an already-trained (infected) model, assumes minimal prior knowledge, and is broadly compatible across architectures while requiring lightweight computation (Min et al., 2023).

However, existing fine-tuning methods typically rely on a small, trusted clean set. In practice, this assumption is fragile. Labeling errors and undetected poisoned samples frequently contaminate the

"clean" curation. Our experiments show that state-of-the-art clean-only fine-tuning often fails to reduce attack success rate without harming benign accuracy even under mild contamination. The failure is structural: optimizing benign loss where triggers are absent can leave the trigger-to-target association intact.

Operational pipelines often include steps such as monitoring and incident response, which can flag a handful of suspicious inputs and a hypothesized target class. Since Neural Cleanse (Wang et al., 2019a) and TABOR (Guo et al., 2020), backdoor detection at the inference stage has been extensively studied (Dong et al., 2021a; Xu et al., 2024a; Hu et al., 2024; Xian et al., 2023). Those proposed detectors can routinely achieve extremely high detection accuracy on commonly used benchmark datasets in the computer vision field. Rather than ignoring these auxiliary signals, we adopt a more reasonable post-training setting: the defender has an infected model $f_\theta$, a small mostly benign pool $\mathcal{D}_b$, and a small quarantined mostly malicious pool $\mathcal{D}_m$ of flagged triggered inputs with the known target class $t$. Both pools may be imperfect with possible contamination.

With this setting, we propose Partition-Losses Fine-Tuning (PL), a simple loss-level modification that jointly minimizes benign loss while maximizing the target-class loss on the triggered inputs, directly unlearning the backdoor. When no triggered examples are available, PL reduces to standard clean-only fine-tuning. PL requires no architectural changes and adds only one extra forward pass per iteration (see Section 3). To summarize, our contributions are:

1. **Realistic post-training setting.** We formalize fine-tuning with two small tuning pools: a mostly benign pool $\mathcal{D}_b$ and a quarantined mostly malicious pool $\mathcal{D}_m$ with known target $t$. Contamination in both pools is allowed and controlled by false-positive and false-negative rates.

2. **Novel post-training defense paradigm: Partition-Losses Fine-Tuning (PL).** To our knowledge, PL is the first fine-tuning defense that directly leverages triggered samples and their target label in a joint objective. It minimizes benign loss while maximizing the target-class loss on $\mathcal{D}_m$. PL explicitly unlearns the trigger-to-target mapping.

3. **Robust empirical performance.** Under the same tuning budget, PL matches or surpasses state-of-the-art clean-only fine-tuning using $2 - 2.5\times$ fewer clean samples. The dominance persists under nonzero contamination in both pools, across datasets and attacks, and remains stable over reasonable choices of PL regularization weight $\alpha$ and data volume.

4. **Practicality.** PL is a plug-in loss modification. It is domain- and architecture-agnostic and adds only a minor per-iteration overhead.

The rest of the paper is organized as follows. First, Section 2 discusses related works. Second, Section 3 introduces our proposed method. Next, Section 4 provides a comprehensive experimental analysis. Finally, Section 5 discusses conclusions and implications.

## 2  RELATED WORK

**Backdoor attacks.** Backdoor attacks preserve a model's intended behavior while injecting an additional association between a trigger and an adversarial behavior. In targeted attacks, any input stamped with the trigger is mapped to a fixed class. This behavior is learned from a poisoned dataset, where the attacker embeds a trigger in a small fraction of training inputs and relabels them to the target class (Carlini & Terzis, 2022; Chen et al., 2017; Goldblum et al., 2022; Gu et al., 2019; Li et al., 2021b; Turner et al., 2019). Despite growing interest in language models (Chen et al., 2021; Cui et al., 2022; Pan et al., 2022; Liu et al., 2022), computer vision is the primary and predominantly focused field of backdoor mechanisms and benchmarks (Wu et al., 2022). In this paper, we focus on image classification, though the proposed defense strategy is agnostic to both input modality and model architecture. Evaluation on other tasks is deferred to future work.

**Backdoor defenses.** Existing defenses span three stages of the pipeline. Pre-training defenses aim to detect or neutralize poisoned samples before training through anomaly detection or trigger localization (Doan et al., 2020; Liu et al., 2017). In-training defenses intervene during optimization via regularization or representation learning-based screening to inhibit backdoor injection as the model learns (Wu et al., 2022). These two families require full access to and modification to the training pipelines, which are computationally expensive (Huang et al., 2022; Li et al., 2021a; Zeng et al., 2021). Post-training defenses operate on the trained (infected) model. General strategies include

trigger inversion (Wang et al., 2019b; 2022; 2023; Xu et al., 2024b), neuron or channel pruning (Liu et al., 2018; Min et al., 2023; Wu & Wang, 2021; Zhu et al., 2023), and fine-tuning (Liu et al., 2018; Wu & Wang, 2021; Zheng et al., 2022; Min et al., 2023; Zeng et al., 2021; Zhu et al., 2023). While inversion and pruning can reduce backdoors, they often degrade benign accuracy and transfer poorly across architectures (Min et al., 2023). Therefore, fine-tuning has become a practical default due to its minimal assumptions and broad compatibility. However, existing fine-tuning approaches exhibit a critical limitation. Although real-time monitoring and incident-response pipelines routinely flag malicious inputs with the target class, current fine-tuning methods ignore these signals and rely only on a small trusted clean set. They could only indirectly suppress the backdoor behavior by optimizing the benign loss. To address this problem, we propose Partition-Losses Fine-Tuning (PL), which incorporates triggered inputs with the target label into a joint objective that maximizes target-class loss while minimizing benign loss. PL naturally reduces to clean-only fine-tuning when no triggered samples are available.

## 3 PARTITION-LOSSES FINE-TUNING (PL)

This section provides a detailed explanation of our proposal, Partition-Losses Fine-Tuning. Figure 1 shows an overview of the method.

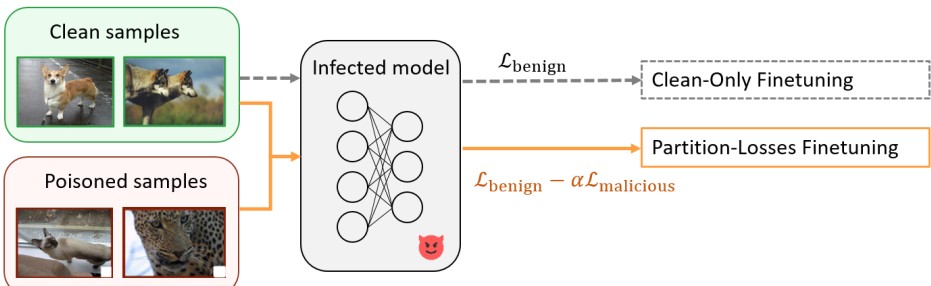

Figure 1: Overview of Partition-Losses Fine-Tuning (PL) (in orange). Compared to clean-only fine-tuning (dashed gray), PL leverages both clean and poisoned samples. It remains effective even in contaminated or diluted scenarios. Note that PL does not use more data: while previous methods assume all fine-tuning samples are clean, we explicitly allow that some portion may be malicious, yet the total number of fine-tuning samples remains comparable to clean-only approaches.

**Problem settings.** We consider an infected model $f_\theta$, where $\theta$ denotes the model parameters, and assume the defender has standard fine-tuning access. Crucially, the defender can run forward or backward passes and update $\theta$. We assume that the defender has two small tuning pools: a benign pool $\mathcal{D}_b$ which contains clean samples and a malicious pool $\mathcal{D}_m$ of triggered samples associated with a known target class $t \in \{1, \ldots, C\}$. Let $\mathcal{P}_{\text{clean}}$ denote the distribution of clean samples $(X, Y)$ and $\mathcal{P}_{\text{trigger}}$ be the distribution of triggered samples $(\tilde{Z}, T)$ with $T \equiv t$.

Both pools may be imperfect. The malicious pool $\mathcal{D}_m$ can include clean samples that are incorrectly flagged as triggered samples (false positives), while the benign pool $\mathcal{D}_b$ can contain triggered samples that are not flagged (false negatives). We denote the contamination rates as follows: $\epsilon_b \in [0, 1]$ is the false negative rate in $\mathcal{D}_b$, which measures the fraction of triggered samples present in the benign pool and $\epsilon_m \in [0, 1]$ is the false positive rate in $\mathcal{D}_m$, which measures the fraction of clean samples in the malicious pool. Therefore, the two tuning pools with some contaminations can be represented as:

$$\mathcal{D}_b^{(\epsilon_b)} = (1 - \epsilon_b)\,\mathcal{P}_{\text{clean}} + \epsilon_b\,\mathcal{P}_{\text{trigger}}, \qquad \mathcal{D}_m^{(\epsilon_m)} = (1 - \epsilon_m)\,\mathcal{P}_{\text{trigger}} + \epsilon_m\,\mathcal{P}_{\text{clean}}.$$

With this setup, we introduce our method, which requires no architectural changes, no knowledge of trigger generation, no access to the original training set, and no clean labels for $\mathcal{D}_m$.

**Proposed method.** In contrast to clean-only fine-tuning methods, which reduce the backdoor indirectly by optimizing on the clean samples, the availability of triggered samples allows us to introduce explicit unlearning signals. Specifically, for a triggered input $\tilde{z}$, we apply gradient ascent on the

target-class cross-entropy. This suppresses the target-class logits and redistributes probability mass to other classes. As a result, the backdoor association is weakened without requiring knowledge of the unknown clean label for $\tilde{z}$.

Therefore, we propose Partition-Losses Fine-Tuning (PL), which leverages clean and triggered data to purify the infected model. PL jointly minimizes the cross-entropy loss on $\mathcal{D}_b$ while maximizing the loss on $\mathcal{D}_m$. Mathematically, we aim to solve the following problem:

$$\min_\theta \mathcal{L}_{\mathrm{PL}}(\theta), \text{ where } \mathcal{L}_{\mathrm{PL}}(\theta) = \mathbb{E}_{(x,y)\sim\mathcal{D}_b^{(\epsilon_b)}}\big[\mathrm{CE}\left(f_\theta(x), y\right)\big] - \alpha\,\mathbb{E}_{(\tilde{z},t)\sim\mathcal{D}_m^{(\epsilon_m)}}\big[\mathrm{CE}\left(f_\theta(\tilde{z}), t\right)\big], \quad (1)$$

with $\alpha > 0$ controlling the trade-off between benign accuracy and backdoor unlearning. A smaller $\alpha$ prioritizes benign accuracy, whereas a larger $\alpha$ promotes backdoor forgetting at the risk of clean accuracy degradation. Compared to clean-only fine-tuning, PL adds one additional forward pass on $\mathcal{D}_m$ per iteration, and the computational cost for this extra step is small. Note that PL reduces exactly to clean-only fine-tuning when no flagged triggered samples are available.

**Optimization procedure.** We keep $\alpha$ fixed throughout fine-tuning. At each tuning iteration, we sample one benign mini-batch and one malicious mini-batch of equal size $n$. Using the class logits $f_\theta(x)$ for clean samples and $f_\theta(\tilde{z})$ for triggered samples, we compute the corresponding cross-entropy losses. For triggered samples, the loss is always evaluated with the target label $t$ rather than its unknown true label. PL then updates $\theta$ using Adam (Kingma & Ba, 2015) on the gradient of the empirical loss $\mathcal{L}_{\mathrm{PL}}$. The complete procedure is summarized in Algorithm 1.

---

**Algorithm 1:** Partition-Losses Fine-Tuning (PL)

---

**Input:** Infected model $f_\theta$; benign pool $\mathcal{D}_b^{(\epsilon_b)}$; malicious pool $\mathcal{D}_m^{(\epsilon_m)}$; target $t$
**Output:** Purified model $f_{\theta^{(I)}}$
**Parameters:** $\alpha > 0$; learning rate $\eta$; tuning iterations $I$; batch size $n$; false negative proportion
$\qquad\qquad$ $\epsilon_b$; false positive proportion $\epsilon_m$
**for** $i = 1$ **to** $I$ **do**
$\quad$ Sample a mini-batch from benign pool $\mathcal{B}_b = \{(x_k, y_k)\}_{k=1}^n \subseteq \mathcal{D}_b^{(\epsilon_b)}$
$\quad$ Calculate cross-entropy loss from benign batch $\mathcal{L}_{\mathrm{Benign}} \leftarrow \frac{1}{n}\sum_{k=1}^n \mathrm{CE}\big(f_\theta(x_k), y_k\big)$
$\quad$ Sample a mini-batch from quarantined malicious pool $\mathcal{B}_m = \{(\tilde{z}_j, t)\}_{j=1}^n \subseteq \mathcal{D}_m^{(\epsilon_m)}$
$\quad$ Calculate cross-entropy loss from malicious batch $\mathcal{L}_{\mathrm{Malicious}} \leftarrow \frac{1}{n}\sum_{j=1}^n \mathrm{CE}\big(f_\theta(\tilde{z}_j), t\big)$
$\quad$ Form PL objective $\mathcal{L}_{\mathrm{PL}} \leftarrow \mathcal{L}_{\mathrm{Benign}} - \alpha\,\mathcal{L}_{\mathrm{Malicious}}$
$\quad$ Update model parameters $\theta^{(i+1)} \leftarrow \theta^{(i)} - \eta\,\nabla_{\theta^{(i)}}\mathcal{L}_{\mathrm{PL}}$
**return** $f_{\theta^{(I)}}$

---

Next, we analyze how optimization of the PL objective influences the attack success rate through its surrogate formulation. The following theorem shows conditions under which PL training provides robustness guarantees over clean-only fine-tuning. The proof is provided in the Appendix A.

**Assumption 1.** *Let $f_\theta : \mathcal{X} \to \Delta^C$ be a classifier with parameters $\theta$, where $\Delta^C$ denotes the probability simplex over $C$ classes. Let $(X, Y) \sim \mathcal{P}_{clean}$ denote a clean input–label pair, and $(\tilde{Z}, T) \sim \mathcal{P}_{trigger}$ a triggered input paired with the fixed target label $T \equiv t$. For contamination levels $\epsilon_b, \epsilon_m \in [0, 1]$, define the (mixture) tuning distributions $\mathcal{D}_b^{\epsilon_b} = (1 - \epsilon_b)\,\mathcal{P}_{clean} + \epsilon_b\,\mathcal{P}_{trigger}, \mathcal{D}_m^{\epsilon_m} = (1 - \epsilon_m)\,\mathcal{P}_{trigger} + \epsilon_m\,\mathcal{P}_{clean}$. The PL objective is*

$$\mathcal{L}_{\mathrm{PL}}(\theta) = \mathbb{E}_{(X,Y)\sim\mathcal{D}_b^{\epsilon_b}}\big[\mathrm{CE}\left(f_\theta(X), Y\right)\big] - \alpha\,\mathbb{E}_{(\tilde{Z},T)\sim\mathcal{D}_m^{\epsilon_m}}\big[\mathrm{CE}\left(f_\theta(\tilde{z}), t\right)\big], \quad \alpha > 0.$$

*The (true) attack success rate is* $\mathrm{ASR}(f_\theta) = \Pr_{(\tilde{Z},T)\sim\mathcal{P}_{trigger}}[f_\theta(\tilde{Z}) = t]$. *Since* $\mathrm{ASR}$ *is non-differentiable, we analyze the surrogate* $\widetilde{\mathrm{ASR}}(f_\theta) = \mathbb{E}_{(\tilde{Z},T)\sim\mathcal{P}_{trigger}}[\log f_\theta(\tilde{Z})_t] = -\mathbb{E}_{(\tilde{Z},T)\sim\mathcal{P}_{trigger}}[\mathrm{CE}(f_\theta(\tilde{Z}), t)]$.

**Theorem 1** (Robustness under imperfect pools via surrogate ASR). *Suppose Assumption 1 holds. Let* $g_{\mathrm{trig}} := \mathbb{E}_{(\tilde{Z},T)\sim\mathcal{P}_{trigger}}\big[\nabla_\theta\,\mathrm{CE}\left(f_\theta(\tilde{Z}), t\right)\big], g_{\mathrm{clean}} := \mathbb{E}_{(X,Y)\sim\mathcal{P}_{clean}}\big[\nabla_\theta\,\mathrm{CE}\left(f_\theta(X), t\right)\big]$. *Assume* $\|g_{\mathrm{clean}}\| \leq \|g_{\mathrm{trig}}\|$. *Then:*

1. (***Relative improvement***) *If $\epsilon_m < \frac{1}{2}$ (i.e., a strict majority of $\mathcal{D}_m^{\epsilon_m}$ are truly triggered), the expected gradient step of (stochastic) gradient descent on $\mathcal{L}_{\text{PL}}$ strictly decreases the surrogate attack success $\widehat{\text{ASR}}(f_\theta)$ more than clean-only fine-tuning, for any $\alpha > 0$.*

2. (***Absolute decrease***) *Moreover, a sufficient condition for $\mathcal{L}_{\text{PL}}$ itself to strictly decrease the surrogate attack success is*

$$\alpha \ > \ \frac{\max\{0, \langle g_{\text{trig}}, \nabla \mathcal{L}_b \rangle\}}{(1 - \epsilon_m)\|g_{\text{trig}}\|^2 + \epsilon_m \langle g_{\text{trig}}, g_{\text{clean}} \rangle}.$$

*In particular, when $\epsilon_m < \frac{1}{2}$, such an $\alpha$ always exists.*

Theorem 1 shows that when the mixed pool is not overly contaminated, the PL objective leverages triggered examples to reduce the surrogate attack success rate more effectively than clean-only fine-tuning. Moreover, with an appropriate choice of $\alpha$, PL training is guaranteed to strictly decrease the surrogate ASR. In other words, as long as the majority of the mixed pool consists of true triggers, PL can systematically suppress the backdoor attack at each optimization step.

## 4 EXPERIMENTS

In the experiments, we aim to answer the following questions: **Q1**. Under ideal partitioning, i.e., $\epsilon_b = \epsilon_m = 0$, how effective is PL in defending against poisoning-based backdoor attacks (Section 4.1)? **Q2**. How robust is PL when fine-tuning with contaminated sets, i.e., when $\epsilon_b > 0$ or $\epsilon_m > 0$ (Section 4.2)? **Q3**. How do different values of $\alpha$ and the amount of available tuning data affect the performance of PL (Section 4.3)?

**Datasets and Models.** We evaluate PL on three widely used benchmark datasets in the backdoor learning literature: CIFAR-10 (Krizhevsky et al., 2009), GTSRB (Stallkamp et al., 2012), and Tiny-ImageNet (Chrabaszcz et al., 2017). Following previous works (Min et al., 2023; Huang et al., 2022; Liu et al., 2018; Nguyen & Tran, 2021; Wu et al., 2022), we use ResNet-18 (He et al., 2016) on CIFAR-10 and GTSRB, and a Swin Transformer (Liu et al., 2021) on Tiny-ImageNet.

**Attack Settings.** To demonstrate the defense effectiveness of PL, we consider five representative dirty-label backdoor attacks from recent works: 1) BadNet (Gu et al., 2019), 2) Blended Attack (Chen et al., 2017), 3) SSBA (Li et al., 2021b), 4) WaNet (Nguyen & Tran, 2021), and 5) Adaptive Blend (Qi et al., 2023). The target label is set to $t = 0$ for all attacks, and the poison rate is 15%. A more detailed experimental setup can be found in Appendix B.

**Defense Baselines.** We compare PL with four tuning-based defenses: vanilla fine-tuning (FT) (Min et al., 2023), fine-tuning with sharpness-aware minimization (FT+SAM) (Zhu et al., 2023), implicit backdoor adversarial unlearning (I-BAU) (Zeng et al., 2021), and feature shift tuning (FST) (Min et al., 2023). The detailed experimental settings are provided in Appendix B.

### 4.1 PL CAN EFFECTIVELY DEFEND AGAINST BACKDOOR ATTACKS

Under the ideal partitioning, where $\epsilon_b = \epsilon_m = 0$, we compare PL with $\alpha = 0.2$ against four clean-only fine-tuning baselines across three datasets and five attacks. We report both attack success rate (ASR) and clean accuracy (C-ACC). We evaluate two tuning-budget regimes: (i) PL-5: PL uses 5% of the training data as the tuning set, while clean-only baselines use 10%; (ii) PL-2: PL uses 2% of the training data as the tuning set, while clean-only baselines use 5%.

Let $N$ be the total number of training samples and set the poison rate $\rho = 0.15$. The tuning set comprises a benign pool $D_b^{\epsilon_b}$ and malicious pool $D_m^{\epsilon_m}$; under ideal partitioning, we have $\epsilon_b = \epsilon_m = 0$. Clean-only baselines exclude poisoned samples from their subsets, so their effective tuning set sizes are $(1 - \rho)\times$ subset sizes. For the total tuning budget, at PL-5, baselines use $1.7\times$ more data than PL ($8.5\%N$ vs $5\%N$); at PL-2, baselines use $2.13 \times$ more data than PL ($4.25\%N$ vs $2\%N$). For the clean sample budget, at PL-5, baselines use $2\times$ more clean data than PL ($8.5\%N$ vs $4.25\%N$); at PL-2, baselines use $2.5\times$ more clean data than PL ($4.25\%N$ vs $1.7\%N$).

Despite having access to substantially fewer clean samples, PL still matches or outperforms clean-only baselines. Table 1a shows that under the PL-5 regime, across all datasets and attack strategies,

PL achieves the lowest ASR (grand mean: 0.004) while maintaining C-ACC (grand mean: 0.784) comparable to the best clean-only baseline. Specifically, on CIFAR-10, PL outperforms all baselines on ASR with a small C-ACC drop compared to the best clean-only method. On GTSRB, PL has zero ASR across all attacks, while preserving C-ACC close to the best baseline. On Tiny-ImageNet, although I-BAU and FST also reduce ASR to zero, they suffer substantial utility loss (mean C-ACC: I-BAU=0.005, FST=0.401). In contrast, PL maintains much higher C-ACC with a mean of 0.693. Meanwhile, FT and FT+SAM maintain high C-ACC but also suffer from very high ASR (mean ASR: FT=0.712, FT+SAM=0.759), leaving the model highly vulnerable. Overall, PL provides the best safety-utility trade-off: near-zero ASR with competitive C-ACC.

Table 1b shows that under the PL-2 regime, PL again achieves the lowest ASR (grand mean ASR=0.005) across all dataset–attack combinations. PL also preserves strong clean accuracy (grand mean C-ACC=0.758), close to the best clean-only baselines. The advantage of PL we observe under the PL-5 regime persists even under a substantially smaller budget, highlighting PL's data efficiency.

Table 1: Performance of PL compared with four baselines under ideal partitioning ($\epsilon_b = \epsilon_m = 0$). Results are reported across three datasets and five attacks. Each entry shows attack success rate (ASR, lower is better) and clean accuracy (C-ACC, higher is better). Bold values indicate the lowest ASR in each row. Panel (a) shows the PL-5 regime, and panel (b) shows the PL-2 regime. PL achieves the lowest ASR and competitive C-ACC compared to baselines in both settings.

| Data | Attack | FST ASR | FST C-ACC | FT ASR | FT C-ACC | I-BAU ASR | I-BAU C-ACC | FT+SAM ASR | FT+SAM C-ACC | PL (Ours) ASR | PL (Ours) C-ACC |
|---|---|---|---|---|---|---|---|---|---|---|---|
| *(a) PL-5: PL uses 5% of the training data vs. baselines use 10% of the training data.* | | | | | | | | | | | |
| | Blended | 0.018 | 0.791 | 0.128 | 0.811 | 0.547 | 0.187 | 0.552 | 0.83 | **0.006** | 0.795 |
| | Adaptive Blend | 0.05 | 0.799 | 0.157 | 0.813 | 0.606 | 0.154 | 0.389 | 0.82 | **0.012** | 0.785 |
| CIFAR-10 | SSBA | 0.039 | 0.78 | 0.041 | 0.806 | 0.795 | 0.147 | 0.079 | 0.81 | **0.013** | 0.774 |
| | BadNet | 0.049 | 0.805 | 0.253 | 0.825 | 0.889 | 0.131 | 0.906 | 0.826 | **0.021** | 0.774 |
| | WaNet | 0.035 | 0.797 | 0.046 | 0.824 | 1 | 0.1 | 0.13 | 0.819 | **0.008** | 0.771 |
| | Mean | 0.038 | 0.794 | 0.125 | 0.816 | 0.767 | 0.144 | 0.411 | 0.821 | **0.012** | 0.78 |
| | Blended | **0** | 0.88 | 0.001 | 0.896 | **0** | 0.142 | 0.013 | 0.914 | **0** | 0.873 |
| | Adaptive Blend | **0** | 0.869 | 0.026 | 0.89 | **0** | 0.191 | 0.419 | 0.91 | **0** | 0.859 |
| GTSRB | SSBA | **0** | 0.874 | 0.003 | 0.895 | 0.007 | 0.197 | 0.005 | 0.905 | **0** | 0.887 |
| | BadNet | **0** | 0.908 | **0** | 0.922 | 0.006 | 0.221 | **0** | 0.918 | **0** | 0.896 |
| | WaNet | 0.001 | 0.895 | 0.015 | 0.906 | 0.006 | 0.09 | 0.076 | 0.904 | **0** | 0.882 |
| | Mean | **0** | 0.885 | 0.009 | 0.902 | 0.004 | 0.168 | 0.103 | 0.91 | **0** | 0.879 |
| | Blended | **0** | 0.412 | 0.996 | 0.717 | **0** | 0.005 | 0.992 | 0.757 | **0** | 0.697 |
| | Adaptive Blend | **0** | 0.412 | 0.996 | 0.717 | **0** | 0.004 | 0.992 | 0.757 | **0** | 0.697 |
| Tiny-ImageNet | SSBA | **0** | 0.387 | 0.972 | 0.703 | **0** | 0.006 | 0.961 | 0.76 | **0** | 0.727 |
| | BadNet | **0** | 0.394 | 0.986 | 0.719 | **0** | 0.005 | 0.993 | 0.766 | **0** | 0.72 |
| | WaNet | 0.005 | 0.399 | 0.468 | 0.702 | **0** | 0.005 | 0.424 | 0.757 | **0** | 0.626 |
| | Mean | 0.001 | 0.401 | 0.884 | 0.712 | **0** | 0.005 | 0.872 | 0.759 | **0** | 0.693 |
| Grand Mean | | 0.013 | 0.693 | 0.339 | 0.81 | 0.257 | 0.106 | 0.462 | 0.83 | **0.004** | 0.784 |
| *(b) PL-2: PL uses 2% of the training data vs. baselines use 5% of the training data.* | | | | | | | | | | | |
| | Blended | 0.013 | 0.795 | 0.143 | 0.81 | 0.635 | 0.213 | 0.649 | 0.82 | **0.006** | 0.789 |
| | Adaptive Blend | 0.049 | 0.794 | 0.159 | 0.806 | 0.054 | 0.131 | 0.357 | 0.817 | **0.017** | 0.79 |
| CIFAR-10 | SSBA | 0.045 | 0.779 | 0.045 | 0.798 | 0.558 | 0.234 | 0.114 | 0.796 | **0.015** | 0.768 |
| | BadNet | 0.036 | 0.795 | 0.237 | 0.817 | 0.571 | 0.23 | 0.917 | 0.82 | **0.025** | 0.77 |
| | WaNet | 0.026 | 0.797 | 0.042 | 0.809 | 0.983 | 0.111 | 0.147 | 0.81 | **0.005** | 0.784 |
| | Mean | 0.034 | 0.792 | 0.125 | 0.808 | 0.56 | 0.184 | 0.437 | 0.813 | **0.014** | 0.78 |
| | Blended | **0** | 0.852 | **0** | 0.877 | **0** | 0.152 | 0.006 | 0.905 | **0** | 0.841 |
| | Adaptive Blend | **0** | 0.836 | 0.003 | 0.869 | **0** | 0.213 | 0.008 | 0.901 | **0** | 0.823 |
| GTSRB | SSBA | **0** | 0.862 | 0.005 | 0.888 | 0.003 | 0.254 | 0.008 | 0.896 | **0** | 0.841 |
| | BadNet | **0** | 0.882 | **0** | 0.905 | 0.106 | 0.247 | **0** | 0.911 | **0** | 0.879 |
| | WaNet | 0.001 | 0.871 | 0.019 | 0.897 | **0** | 0.037 | 0.134 | 0.895 | **0** | 0.841 |
| | Mean | **0** | 0.861 | 0.005 | 0.887 | 0.022 | 0.181 | 0.031 | 0.902 | **0** | 0.845 |
| | Blended | **0** | 0.295 | 0.999 | 0.721 | 0.051 | 0.579 | 0.752 | 0.74 | **0** | 0.66 |
| | Adaptive Blend | **0** | 0.295 | 0.999 | 0.721 | 0.051 | 0.579 | 0.752 | 0.74 | **0** | 0.663 |
| Tiny-ImageNet | SSBA | **0** | 0.309 | 0.954 | 0.706 | **0** | 0.005 | 0.944 | 0.745 | **0** | 0.65 |
| | BadNet | **0** | 0.285 | 0.965 | 0.72 | **0** | 0.005 | 0.99 | 0.748 | **0** | 0.656 |
| | WaNet | 0.003 | 0.29 | 0.946 | 0.715 | **0** | 0.005 | 0.71 | 0.744 | **0** | 0.618 |
| | Mean | 0.001 | 0.295 | 0.973 | 0.717 | 0.02 | 0.235 | 0.83 | 0.743 | **0** | 0.649 |
| Grand Mean | | 0.012 | 0.649 | 0.368 | 0.804 | 0.201 | 0.2 | 0.433 | 0.819 | **0.005** | 0.758 |

## 4.2 PL IS ROBUST TO CONTAMINATION IN FINE-TUNING DATASETS

The "benign" tuning pool may contain triggered samples for two main reasons. First, crowd-sourced or large-scale annotation pipelines can introduce inconsistent labels, naturally leading to mislabeled samples (Northcutt et al., 2021). Second, because backdoor defenses and attacks co-evolve, we may not have the optimal screener that generalizes universally, leaving some triggered samples undetected (Dong et al., 2021b; Hayase et al., 2021). The malicious pool may also include some clean samples. Therefore, collecting perfectly clean or poisoned fine-tuning datasets is often infeasible.

To test the robustness of PL under more realistic conditions, we evaluate the same two tuning regimes as in Section 4.1: PL-5 (PL uses $5\%$ of the training data vs. baselines use $10\%$ of the training data) and PL-2 (PL uses $2\%$ of the training data vs. baselines use $5\%$ of the training data). Let $N_{tune} = |\mathcal{D}_b^{\epsilon_b}| + |\mathcal{D}_m^{\epsilon_m}|$ be the total number of fine-tuning samples, which consists of the benign pool $D_b^{\epsilon_b}$ and malicious pool $D_m^{\epsilon_m}$. Under each regime, we vary contamination in the two pools and evaluate three cases: (a) contamination in the benign pool ($\epsilon_b = 0.1$, $\epsilon_m = 0$), (b) contamination in the malicious pool ($\epsilon_b = 0$, $\epsilon_m = 0.1$), and (c) contamination in both pools $\epsilon_b = 0.1$, $\epsilon_m = 0.1$. Results are shown in Table 2 and Table 3 (Appendix C).

**Contamination in the benign pool** ($\epsilon_b = 0.1$, $\epsilon_m = 0$): Table 2a shows that under the PL-2 regime, *PL preserves the best safety–utility trade-off*, achieving low ASR (grand mean: 0.107) with competitive C-ACC (0.760). In comparison, baselines fail to remove the backdoor while maintaining functionality. For example, I-BAU reaches near-zero ASR on the GTSRB dataset in four out of five attacks but spikes to one for the remaining attack. On CIFAR-10 and Tiny-ImageNet, its ASR spans from near-zero to near-one while substantially sacrificing C-ACC (grand mean: 0.132). FT+SAM maintains the highest utility (C-ACC grand mean: 0.818), leaving the model infected (ASR grand mean: 0.787). FST and FT have moderately high ASR on CIFAR-10 and GTSRB, yet reach near-one ASR on Tiny-ImageNet.

**Contamination in the malicious pool** ($\epsilon_b = 0$, $\epsilon_m = 0.1$): We perturb the data by moving $\epsilon_m$ fraction of clean samples into the malicious pool. This reduces the benign pool available to the clean-only baselines and dilutes PL's malicious pool. Under this setting, Table 2b shows that PL again provides the strongest defense, with the lowest ASR (grand mean: 0.002) and competitive utility (C-ACC grand mean: 0.656). FT and FT+SAM achieve the highest utility (C-ACC grand mean: 0.809 and 0.819, respectively), but at the cost of high ASR (grand means: 0.389 and 0.512). I-BAU produces low ASR in some cases at the cost of utility (C-ACC grand mean: 0.133). FST provides a more balanced performance (grand mean ASR 0.017, C-ACC 0.653).

**Contamination in both benign and malicious pools** ($\epsilon_b = 0.1$, $\epsilon_m = 0.1$): This is the most challenging yet most realistic setting: the benign pool contains triggered samples while the malicious pool is diluted with clean samples. Therefore, clean-only baselines face fewer and impure clean samples, and PL has to deal with conflicts in both benign and malicious tuning sets. Despite these challenges, PL consistently achieves near-zero ASR (grand mean: 0.010) while preserving reasonable utility (C-ACC grand mean: 0.661) as shown in Table 2c. In contrast, FT, FT+SAM, and FST maintain high utility (C-ACC grand means: 0.807, 0.857, and 0.656) but leave the model heavily poisoned (ASR grand means: 0.618, 0.857, and 0.523). I-BAU performs worst overall in this setting, returning extremely low C-ACC (grand mean: 0.192) and high ASR (grand mean: 0.615).

Across all contamination settings under the PL-2 regime, our method PL consistently achieves the lowest ASR with competitive C-ACC, demonstrating robustness even when both tuning pools are corrupted. More importantly, it does so while using less than half the tuning data (PL: $2\%N$ vs. baseline: $4.25\%N$). Similar results hold under the PL-5 regime in Table 3 (Appendix C).

## 4.3 ABLATION STUDIES FOR PL

In this section, we conduct ablation studies for PL concerning the trade-off parameter $\alpha$ and the fine-tuning dataset's size.

### 4.3.1 SENSITIVITY ANALYSIS ON PL TRADE-OFF PARAMETER $\alpha$

We study the stability of PL under different $\alpha$ values by evaluating $\alpha \in \{0.1, 0.2, 0.3, 0.4, 0.5\}$ on the CIFAR-10 dataset under both the perfect partition ($\epsilon_b = \epsilon_m = 0$) and the realistic contamination

Table 2: The performance of PL was compared with four baselines on contaminated fine-tuning data under the PL-2 regime and evaluated under three contamination settings. Bold values indicate the best performance (lowest ASR). PL consistently outperforms baselines on the safety-utility trade-off across all three settings.

| | | FST | | FT | | I-BAU | | FT+SAM | | PL (Ours) | |
|---|---|---|---|---|---|---|---|---|---|---|---|
| Data | Attack | ASR | C-ACC | ASR | C-ACC | ASR | C-ACC | ASR | C-ACC | ASR | C-ACC |
| *(a) Benign pool contamination $\epsilon_b = 0.1$, $\epsilon_m = 0$* | | | | | | | | | | | |
| CIFAR-10 | Blended | 0.225 | 0.787 | 0.336 | 0.809 | 0.344 | 0.264 | 0.885 | 0.818 | **0.061** | 0.789 |
| | Adaptive Blend | 0.245 | 0.797 | 0.339 | 0.808 | 0.562 | 0.159 | 0.879 | 0.811 | **0.091** | 0.787 |
| | SSBA | 0.075 | 0.769 | 0.063 | 0.801 | 0.694 | 0.16 | 0.221 | 0.794 | **0.018** | 0.773 |
| | BadNet | 0.205 | 0.793 | 0.349 | 0.815 | 0.808 | 0.232 | 0.957 | 0.823 | **0.057** | 0.773 |
| | WaNet | 0.092 | 0.798 | 0.128 | 0.812 | 0.999 | 0.102 | 0.507 | 0.802 | **0.033** | 0.78 |
| | Mean | 0.168 | 0.789 | 0.243 | 0.809 | 0.681 | 0.183 | 0.69 | 0.81 | **0.052** | 0.78 |
| GTSRB | Blended | 0.363 | 0.875 | 0.481 | 0.89 | **0** | 0.127 | 0.939 | 0.907 | 0.324 | 0.839 |
| | Adaptive Blend | 0.382 | 0.853 | 0.505 | 0.878 | **0** | 0.157 | 0.927 | 0.9 | 0.37 | 0.822 |
| | SSBA | 0.37 | 0.882 | 0.536 | 0.9 | **0.013** | 0.189 | 0.93 | 0.9 | 0.298 | 0.86 |
| | BadNet | 0.318 | 0.88 | 0.524 | 0.9 | **0.091** | 0.114 | 0.936 | 0.905 | 0.299 | 0.868 |
| | WaNet | 0.107 | 0.875 | 0.181 | 0.897 | 0.998 | 0.006 | 0.625 | 0.899 | **0.058** | 0.844 |
| | Mean | 0.308 | 0.873 | 0.445 | 0.893 | **0.22** | 0.119 | 0.871 | 0.902 | 0.27 | 0.847 |
| Tiny-ImageNet | Blended | 1 | 0.32 | 1 | 0.722 | 1 | 0.014 | 1 | 0.744 | **0** | 0.677 |
| | Adaptive Blend | 1 | 0.32 | 1 | 0.722 | 1 | 0.014 | 1 | 0.744 | **0** | 0.665 |
| | SSBA | 0.994 | 0.307 | 0.998 | 0.718 | 0.996 | 0.428 | 0.997 | 0.749 | **0** | 0.654 |
| | BadNet | 0.993 | 0.326 | 0.995 | 0.724 | 0.99 | 0.013 | 0.995 | 0.751 | **0** | 0.651 |
| | WaNet | 0.991 | 0.333 | 0.992 | 0.702 | **0** | 0.005 | 0.001 | 0.726 | **0** | 0.625 |
| | Mean | 0.996 | 0.321 | 0.997 | 0.718 | 0.797 | 0.095 | 0.799 | 0.743 | **0** | 0.654 |
| Grand Mean | | 0.491 | 0.661 | 0.562 | 0.807 | 0.566 | 0.132 | 0.787 | 0.818 | **0.107** | 0.76 |
| *(b) Malicious pool contamination $\epsilon_b = 0$, $\epsilon_m = 0.1$* | | | | | | | | | | | |
| CIFAR-10 | Blended | 0.033 | 0.805 | 0.218 | 0.818 | 0.66 | 0.2 | 0.629 | 0.818 | **0** | 0.757 |
| | Adaptive Blend | 0.088 | 0.801 | 0.252 | 0.814 | 0.349 | 0.25 | 0.514 | 0.818 | **0.001** | 0.758 |
| | SSBA | 0.035 | 0.784 | 0.054 | 0.802 | 0.373 | 0.306 | 0.147 | 0.797 | **0.006** | 0.745 |
| | BadNet | 0.073 | 0.808 | 0.27 | 0.819 | 0.643 | 0.325 | 0.637 | 0.822 | **0.007** | 0.74 |
| | WaNet | 0.025 | 0.799 | 0.051 | 0.812 | 1 | 0.102 | 0.196 | 0.807 | **0.004** | 0.759 |
| | Mean | 0.051 | 0.799 | 0.169 | 0.813 | 0.605 | 0.237 | 0.425 | 0.812 | **0.004** | 0.752 |
| GTSRB | Blended | **0** | 0.881 | **0** | 0.897 | **0** | 0.159 | **0** | 0.905 | 0.001 | 0.748 |
| | Adaptive Blend | **0** | 0.859 | **0** | 0.892 | **0** | 0.142 | 0.001 | 0.897 | **0** | 0.697 |
| | SSBA | **0** | 0.866 | **0** | 0.893 | 0.001 | 0.167 | **0** | 0.892 | **0** | 0.631 |
| | BadNet | **0** | 0.889 | 0.013 | 0.907 | **0** | 0.286 | 0.51 | 0.904 | 0.004 | 0.701 |
| | WaNet | 0.001 | 0.882 | 0.035 | 0.896 | **0** | 0.034 | 0.17 | 0.895 | **0** | 0.731 |
| | Mean | **0** | 0.875 | 0.01 | 0.897 | **0** | 0.158 | 0.136 | 0.899 | 0.001 | 0.702 |
| Tiny-ImageNet | Blended | **0** | 0.282 | 0.999 | 0.714 | **0** | 0.006 | 0.999 | 0.743 | **0** | 0.476 |
| | Adaptive Blend | **0** | 0.282 | 0.999 | 0.714 | **0** | 0.006 | 0.999 | 0.743 | **0** | 0.496 |
| | SSBA | **0** | 0.289 | 0.976 | 0.717 | **0** | 0.005 | 0.966 | 0.744 | **0** | 0.547 |
| | BadNet | **0** | 0.287 | 0.985 | 0.714 | **0** | 0.005 | 0.992 | 0.751 | **0** | 0.53 |
| | WaNet | **0** | 0.287 | 0.981 | 0.727 | **0** | 0.005 | 0.918 | 0.743 | **0** | 0.527 |
| | Mean | **0** | 0.285 | 0.988 | 0.717 | **0** | 0.005 | 0.975 | 0.745 | **0** | 0.515 |
| Grand Mean | | 0.017 | 0.653 | 0.389 | 0.809 | 0.202 | 0.133 | 0.512 | 0.819 | **0.002** | 0.656 |
| *(c) Both pools contamination $\epsilon_b = 0.1$, $\epsilon_m = 0.1$* | | | | | | | | | | | |
| CIFAR-10 | Blended | 0.333 | 0.801 | 0.475 | 0.819 | 0.751 | 0.15 | 0.888 | 0.821 | **0.049** | 0.716 |
| | Adaptive Blend | 0.282 | 0.796 | 0.425 | 0.812 | 0.217 | 0.218 | 0.873 | 0.814 | **0.033** | 0.74 |
| | SSBA | 0.088 | 0.783 | 0.105 | 0.797 | 0.418 | 0.258 | 0.273 | 0.796 | **0.012** | 0.739 |
| | BadNet | 0.323 | 0.798 | 0.539 | 0.816 | 0.772 | 0.181 | 0.955 | 0.822 | **0.004** | 0.722 |
| | WaNet | 0.126 | 0.794 | 0.179 | 0.812 | 1 | 0.1 | 0.493 | 0.8 | **0.026** | 0.746 |
| | Mean | 0.23 | 0.794 | 0.345 | 0.811 | 0.632 | 0.181 | 0.696 | 0.811 | **0.025** | 0.733 |
| GTSRB | Blended | 0.462 | 0.875 | 0.587 | 0.894 | **0.004** | 0.197 | 0.921 | 0.902 | 0.006 | 0.729 |
| | Adaptive Blend | 0.413 | 0.875 | 0.549 | 0.895 | 0.028 | 0.193 | 0.938 | 0.902 | 0.01 | 0.76 |
| | SSBA | 0.308 | 0.883 | 0.57 | 0.9 | **0.001** | 0.231 | 0.952 | 0.893 | 0.002 | 0.756 |
| | BadNet | 0.344 | 0.881 | 0.607 | 0.897 | 0.609 | 0.205 | 0.939 | 0.902 | **0.006** | 0.695 |
| | WaNet | 0.187 | 0.874 | 0.236 | 0.895 | 0.481 | 0.014 | 0.632 | 0.893 | **0.004** | 0.754 |
| | Mean | 0.343 | 0.878 | 0.51 | 0.896 | 0.225 | 0.168 | 0.876 | 0.898 | **0.006** | 0.739 |
| Tiny-ImageNet | Blended | 1 | 0.307 | 1 | 0.704 | 0.984 | 0.36 | 1 | 0.738 | **0** | 0.512 |
| | Adaptive Blend | 1 | 0.307 | 1 | 0.704 | 0.984 | 0.36 | 1 | 0.738 | **0** | 0.493 |
| | SSBA | 0.998 | 0.284 | 0.999 | 0.722 | 0.999 | 0.393 | 0.999 | 0.746 | **0** | 0.557 |
| | BadNet | 0.995 | 0.297 | 0.995 | 0.716 | 0.976 | 0.011 | 0.994 | 0.739 | **0** | 0.537 |
| | WaNet | 0.991 | 0.286 | 0.997 | 0.723 | 1 | 0.005 | 0.995 | 0.744 | **0.002** | 0.462 |
| | Mean | 0.997 | 0.296 | 0.998 | 0.714 | 0.989 | 0.226 | 0.998 | 0.741 | **0** | 0.512 |
| Grand Mean | | 0.523 | 0.656 | 0.618 | 0.807 | 0.615 | 0.192 | 0.857 | 0.817 | **0.01** | 0.661 |

in both pools ($\epsilon_b = \epsilon_m = 0.1$). Figure 2 shows consistent patterns across five attacks in both settings: (i) ASR decreases as $\alpha$ increases; (ii) C-ACC decays gradually as $\alpha$ grows. (iii) $\alpha$ balances the safety–utility trade-off. Overall, PL does not show oscillatory behavior over the grid and supports a single default choice (we use $\alpha = 0.2$ in Section 4.1 and 4.2) without per-attack tuning.

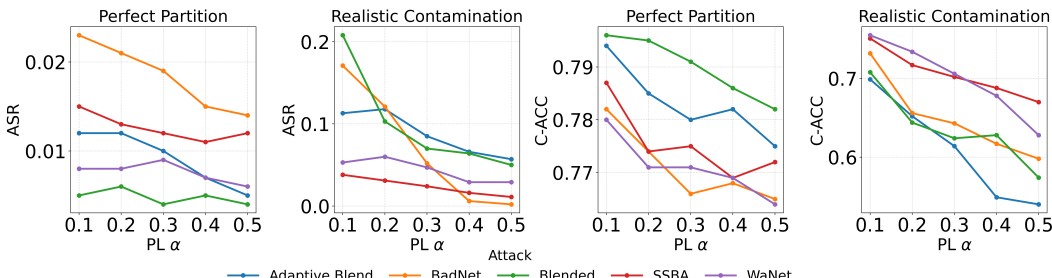

Figure 2: Sensitivity of PL to the trade-off parameter $\alpha$ on CIFAR-10. Results are shown under perfect partition and realistic contamination. Increasing $\alpha$ consistently reduces ASR at a modest cost to C-ACC across five attacks. Within the test $\alpha$ range, PL maintains low ASR while preserving reasonable utility under both perfect partition and realistic contamination.

### 4.3.2 SENSITIVITY ANALYSIS ON TUNING DATASET SIZE

In addition, we study the impact of the fine-tuning data availability by varying tuning fraction of the training set. We use the fraction $\{2\%, 5\%, 10\%\}$ under two conditions: (i) perfect partition ($\epsilon_b = \epsilon_m = 0$) and (ii) realistic contamination in both pools ($\epsilon_b = \epsilon_m = 0.1$). Figure 3 shows that PL consistently achieves the lowest ASR across tuning budgets (purple line) while maintaining C-ACC close to the best clean-only baselines in both settings. Increasing the fraction of the tuning data generally improves performance for all methods, but PL achieves strong safety–utility trade-offs even at the smallest budget, highlighting its data efficiency.

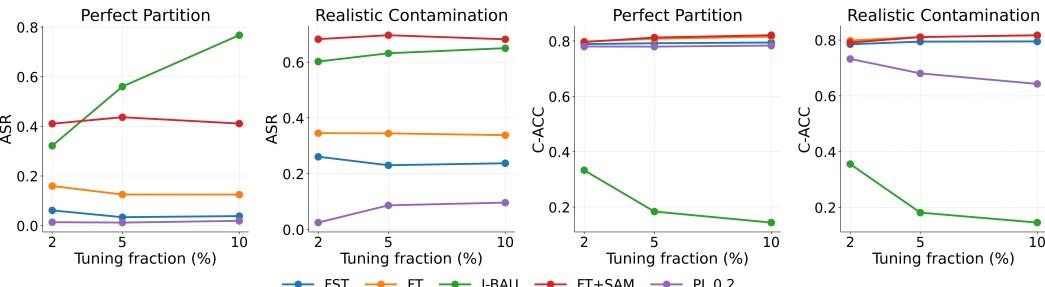

Figure 3: Sensitivity of PL and baselines to tuning data availability on CIFAR-10. Results are shown under perfect partition and realistic contamination, with different per-pool tuning fractions. PL consistently achieves the lowest attack success rate (ASR) while maintaining clean accuracy (C-ACC), which is close to the best clean-only baselines, showing its data efficiency across budgets.

## 5 CONCLUSION

This paper proposes Partition-Losses Fine-Tuning (PL), a new defense method for mitigating poisoning-based backdoor attacks. Unlike existing clean-only baselines, which rely on the unrealistic assumption of perfectly clean tuning data, PL explicitly incorporates a small set of flagged malicious samples during fine-tuning. It does so by minimizing the benign loss while simultaneously maximizing the malicious loss, providing a direct unlearning signal for the backdoor. Extensive experiments across multiple datasets and attacks show that PL uses fewer tuning samples than existing baselines, yet consistently achieves the lowest attack success rate while maintaining competitive clean accuracy. These results hold under both ideal partitioning and realistic contamination, and sensitivity analyses confirm PL's stability across a wide range of hyperparameter values and data budgets.

This work broadens the design space of practical backdoor defenses by showing that quarantined malicious samples can be effectively leveraged rather than discarded. The proposed method is simple, model-agnostic, and data-efficient, making it well-suited for deployment in real-world settings

where annotation noise and undetected triggers are unavoidable. In the broader context, PL provides a step toward building more trustworthy and resilient machine learning systems, with potential benefits for safety-critical applications such as healthcare, autonomous driving, and cybersecurity.

ETHICS STATEMENT

This work studies defenses against poisoning-based backdoor attacks in deep learning models. We propose Partition-Losses Fine-Tuning (PL), a defense method to purify an infected model. No human subjects or personally identifiable information are involved. We do not foresee ethical concerns related to fairness, privacy, or confidentiality beyond standard considerations for public benchmark datasets, and we comply with the ICLR Code of Ethics.

REPRODUCIBILITY STATEMENT

We provide a complete description of our method in Section 3. Details of hyperparameters, data splits, and evaluation metrics are given in Section 4 and Appendix B. All experiments are conducted on publicly available benchmark datasets (CIFAR-10, GTSRB, and Tiny-ImageNet). We will release code upon acceptance to facilitate replication.

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

## A    PROOF FOR THEOREM 1

*Proof.* The expected gradient of $\mathcal{L}_{\text{PL}}$ is

$$\nabla_\theta \mathcal{L}_{\text{PL}} = \underbrace{\mathbb{E}_{(X,Y)\sim\mathcal{D}_b^{\epsilon_b}}\big[\nabla_\theta \text{CE}(f_\theta(X), Y)\big]}_{=:\nabla\mathcal{L}_b} - \alpha \underbrace{\mathbb{E}_{(\tilde{Z},T)\sim\mathcal{D}_m^{\epsilon_m}}\big[\nabla_\theta \text{CE}(f_\theta(\tilde{Z}), t)\big]}_{=:\nabla\mathcal{L}_m}. \tag{2}$$

Note that

$$\nabla\mathcal{L}_m = (1 - \epsilon_m)\mathbb{E}_{\mathcal{P}_{\text{trigger}}}\big[\nabla CE(f_\theta(\tilde{Z}), t)\big] + \epsilon_m \mathbb{E}_{\mathcal{P}_{\text{clean}}}\big[\nabla CE(f_\theta(X), t)\big]$$
$$= (1 - \epsilon_m)g_{\text{trigger}} + \epsilon_m g_{\text{clean}}. \tag{3}$$

For a small step size $\eta > 0$, the parameter update is $\theta^+ = \theta - \eta\nabla_\theta L(\theta)$. The change in surrogate ASR is $\Delta\widetilde{\text{ASR}} = \widetilde{\text{ASR}}(f_{\theta+}) - \widetilde{\text{ASR}}(f_\theta)$. By a first-order Taylor expansion of $\widetilde{\text{ASR}}$ around $\theta$,

$$\Delta\widetilde{\text{ASR}} \approx \big\langle\nabla_\theta\widetilde{\text{ASR}}(f_\theta), \theta^+ - \theta\big\rangle.$$

Substituting the update rule gives

$$\Delta\widetilde{\text{ASR}} \approx -\eta\big\langle\nabla_\theta\widetilde{\text{ASR}}(f_\theta), \nabla_\theta L(\theta)\big\rangle. \tag{4}$$

Since $\widetilde{\text{ASR}}(f_\theta) = -\mathbb{E}_{\mathcal{P}_{\text{trigger}}}\big[\text{CE}(f_\theta(\tilde{Z}), t)\big]$, we have $\nabla_\theta\widetilde{\text{ASR}}(f_\theta) = -g_{\text{trig}}$. Equation 4 simplifies to

$$\Delta\widetilde{\text{ASR}} \approx \eta\big\langle g_{\text{trig}}, \nabla_\theta L(\theta)\big\rangle.$$

Substituting equation 2 and equation 3 gives

$$\Delta\widetilde{\text{ASR}} \approx \eta\Big(\langle g_{\text{trig}}, \nabla\mathcal{L}_b\rangle - \alpha(1 - \epsilon_m)\|g_{\text{trig}}\|^2 - \alpha\,\epsilon_m\langle g_{\text{trig}}, g_{\text{clean}}\rangle\Big).$$

By comparison, clean-only fine-tuning corresponds to $\alpha = 0$, in which case

$$\Delta\widetilde{\text{ASR}}_{\text{clean}} \approx \eta\langle g_{\text{trig}}, \nabla\mathcal{L}_b\rangle.$$

Therefore, the difference between PL and clean-only is

$$\Delta\widetilde{\text{ASR}} - \Delta\widetilde{\text{ASR}}_{\text{clean}} \approx -\eta\alpha\Big((1 - \epsilon_m)\|g_{\text{trig}}\|^2 + \epsilon_m\langle g_{\text{trig}}, g_{\text{clean}}\rangle\Big). \tag{5}$$

By Cauchy–Schwarz and the assumption $\|g_{\text{clean}}\| \leq \|g_{\text{trig}}\|$, we have

$$\langle g_{\text{trig}}, g_{\text{clean}}\rangle \geq -\|g_{\text{trig}}\|\,\|g_{\text{clean}}\| \geq -\|g_{\text{trig}}\|^2.$$

Therefore the bracketed term in equation 5 admits the *lower* bound

$$(1 - \epsilon_m)\|g_{\text{trig}}\|^2 + \epsilon_m\langle g_{\text{trig}}, g_{\text{clean}}\rangle \geq (1 - \epsilon_m)\|g_{\text{trig}}\|^2 - \epsilon_m\|g_{\text{trig}}\|^2$$
$$= (1 - 2\epsilon_m)\,\|g_{\text{trig}}\|^2.$$

If $\epsilon_m < \frac{1}{2}$, the right-hand side is strictly positive. Plugging this into equation 5 yields

$$\Delta\widetilde{\text{ASR}} - \Delta\widetilde{\text{ASR}}_{\text{clean}} \approx -\eta\alpha\Big((1 - \epsilon_m)\|g_{\text{trig}}\|^2 + \epsilon_m\langle g_{\text{trig}}, g_{\text{clean}}\rangle\Big) < 0,$$

This shows that the PL update always reduces the surrogate ASR *more* than clean-only fine-tuning whenever $\epsilon_m < \frac{1}{2}$, for any $\alpha > 0$.

Finally, to obtain an *absolute* decrease guarantee, note from

$$\Delta\widetilde{\text{ASR}} \approx \eta\Big(\langle g_{\text{trig}}, \nabla\mathcal{L}_b\rangle - \alpha(1 - \epsilon_m)\|g_{\text{trig}}\|^2 - \alpha\epsilon_m\langle g_{\text{trig}}, g_{\text{clean}}\rangle\Big)$$

that a sufficient condition for $\Delta\widetilde{\text{ASR}} < 0$ is

$$\alpha > \frac{\max\{0, \langle g_{\text{trig}}, \nabla\mathcal{L}_b\rangle\}}{(1 - \epsilon_m)\|g_{\text{trig}}\|^2 + \epsilon_m\langle g_{\text{trig}}, g_{\text{clean}}\rangle}.$$

In particular, when $\epsilon_m < \frac{1}{2}$, such an $\alpha$ always exists. $\qquad\square$

## B    EXPERIMENTAL SETUP

**Datasets and Models.** Following previous works (Min et al., 2023; Huang et al., 2022; Liu et al., 2018; Nguyen & Tran, 2021; Wu et al., 2022), we evaluate on three widely used benchmark datasets in the backdoor learning literature: CIFAR-10, GTSRB, and Tiny-ImageNet. CIFAR-10 and GT-SRB contain images of $32 \times 32$ resolution of 10 and 43 categories. We use ResNet-18 to build a backdoor model for those two datasets. Tiny-ImageNet contains images of $64 \times 64$ resolution and 100 categories. They are resized to $224 \times 224$. We used a pre-trained SwinTransformer provided by PyTorch to implement the backdoor.

**Attack Settings.** We conducted all the experiments with $4 \times$ NVIDIA RTX A6000 GPUs (48 GiB each). We implement five representative backdoor attacks from recent works: BadNet, Blended, SSBA, WaNet, and Adaptive Blend. For BadNet (Gu et al., 2019), we use a white square as the backdoor trigger and stamp the pattern at the lower right corner of the image; for Blended (Chen et al., 2017), we adopt the uniform noise as the trigger and set the blend ratio as 0.1 for both the training and inference phases; for WaNet (Nguyen & Tran, 2021), we set the size of the backward warping field as four and the strength of the wrapping field as 0.5; SSBA (Li et al., 2021b) is a dynamic invisible-trigger attack. We implement the complete SSBA encoder–decoder pipeline: we train the SSBA steganographic trigger, where a learnable encoder injects a binary fingerprint into each input image, and a decoder is jointly trained to recover this fingerprint at a poisoning rate of 0.15; Adaptive Blend (Qi et al., 2023) is an enhanced variant of the classical blended attack. It consists of input-dependent partial blending and probabilistic label flipping. During poisoning, the image is divided into a grid of patches; only half of the patches are randomly selected for blending with the trigger at the poisoning rate of 0.15. The remaining patches stay unchanged, creating a spatially sparse and input-dependent backdoor pattern.

All backdoor models are trained for 50 epochs with an initial learning rate of 0.0001. Then, we fine-tune each model for 10 epochs.

**Defenses Settings.** We compare four tuning-based defenses and one extra state-of-the-art defense strategy, I-BAU. For tuning-based defenses, we mainly consider FT-init, FT+SAM, and FST. For all the defense settings, set the batch size as 128 on CIFAR-10 and GTSRB, and set the batch size as 32 on Tiny-ImageNet due to the memory limit. I-BAU utilizes the implicit hypergradient to account for the interdependence between inner and outer optimization (Zeng et al., 2021). FT-Init randomly re-initializes the linear head and fine-tunes the whole model architecture (Min et al., 2023). FT+SAM replace the optimizer in FT-Init with SAM (Min et al., 2023; Zhu et al., 2023). FST shifts features by encouraging the discrepancy between the tuned classifier weight and the original backdoored classifier weight (Min et al., 2023).

## C    PERFORMANCE UNDER CONTAMINATED PARTITION OF TUNING SETS.

**Contamination in the benign pool** ($\epsilon_b = 0.1$, $\epsilon_m = 0$)**:** Table 3a shows that under the PL-5 regime, *PL preserves the best safety–utility trade-off*, achieving low ASR (grand mean: 0.063) with competitive C-ACC (0.784). In comparison, baselines fail to remove the backdoor while maintaining functionality. For example, I-BAU reaches near-zero ASR on the GTSRB dataset in four out of five attacks but spikes to around 0.9 for the remaining attack. On CIFAR-10 and Tiny-ImageNet, its ASR spans from near-zero to near-one while substantially sacrificing C-ACC (grand mean: 0.104). FT+SAM maintains the highest utility (C-ACC grand mean: 0.827), leaving the model infected (ASR grand mean: 0.791). FST and FT have moderately high ASR on CIFAR-10 and GTSRB, yet reach near-one ASR on Tiny-ImageNet.

**Contamination in the malicious pool** ($\epsilon_b = 0$, $\epsilon_m = 0.1$)**:** We perturb the data by moving $\epsilon_m$ fraction of clean samples into the malicious pool. This reduces the benign pool available to the clean-only baselines and dilutes PL's malicious pool. Under this setting, Table 3b shows that PL again provides the strongest defense, with the lowest ASR (grand mean: 0.002) and competitive utility (C-ACC grand mean: 0.598). FT and FT+SAM achieve the highest utility (C-ACC grand mean: 0.813 and 0.828, respectively), but at the cost of high ASR (grand means: 0.363 and 0.513). I-BAU produces low ASR in some cases at the cost of utility (C-ACC grand mean: 0.092). FST provides a more balanced performance (grand mean ASR 0.015, C-ACC 0.693).

**Contamination in both benign and malicious pools** ($\epsilon_b = 0.1$, $\epsilon_m = 0.1$): This is the most challenging yet most realistic setting: the benign pool contains triggered samples while the malicious pool is diluted with clean samples. Therefore, clean-only baselines face fewer and impure clean samples, and PL has to deal with conflicts in both benign and malicious tuning sets. Despite these challenges, PL consistently achieves near-zero ASR (grand mean: 0.034) while preserving reasonable utility (C-ACC grand mean: 0.615) as shown in Table 3c. In contrast, FT, FT+SAM, and FST maintain high utility (C-ACC grand means: 0.815, 0.829, and 0.7) but leave the model heavily poisoned (ASR grand mean: 0.639, 0.797, and 0.548). I-BAU performs worst overall in this setting, returning extremely low C-ACC (grand mean: 0.099) and high ASR (grand mean: 0.797).

Across all contamination settings under the PL-5 regime, our method PL consistently achieves the lowest ASR with competitive C-ACC, demonstrating robustness even when both tuning pools are corrupted. More importantly, it does so while using around half of the tuning data compared to the baselines (Baseline $8.5\%N$ vs PL $5\%N$).

# D ADDITIONAL ABLATION: SENSITIVITY TO PARTITION CONTAMINATION

We evaluate both the proposed Partition-Loss fine-tuning (PL) and the baseline fine-tuning method under controlled benign contamination (False Negative, FN only), malicious contamination (False Positive, FP only), and combined (FN+FP) contamination at the rate of 5%, 10%, 15%, 20%. These results (Figures 4, 5, 6, 7, 8, 9) complement the main text and demonstrate the expected performance degradation as contamination increases, while also showing that PL remains competitive under all tested regimes.

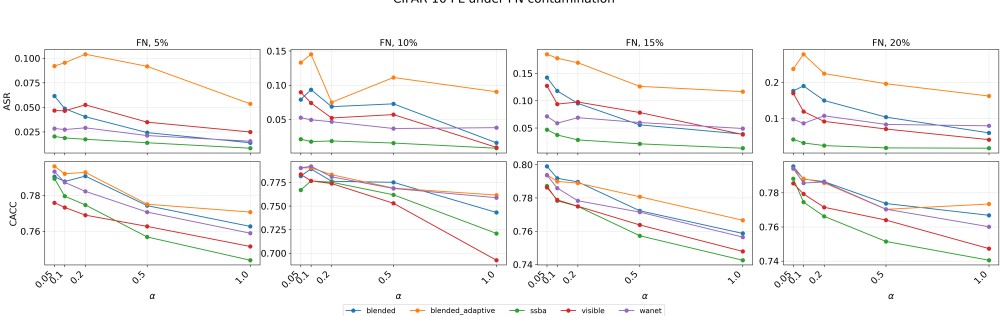

Figure 4: **PL under benign contamination (FN only).**

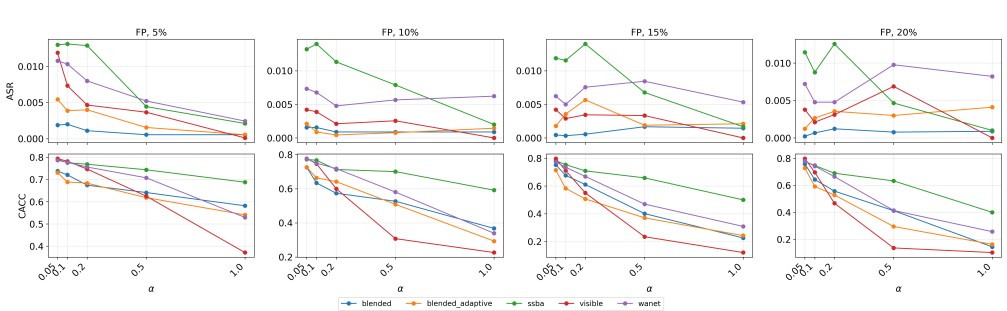

Figure 5: **PL under malicious contamination (FP only).**

# E ADDITIONAL ABLATION: GENERALIZATION TO UNKNOWN ATTACKS

The cross-attack fine-tuning tables 4 and 5 show: under perfect partition and benign contamination, clean accuracy remains high and stable across all unseen attacks. ASR remains low when PL

Table 3: Performance of PL compared with four baselines under contaminated fine-tuning data under the PL-5 regime, evaluated under three contamination settings. Bold values indicate the best performance (lowest ASR or highest C-ACC). PL consistently outperforms baselines on the safety-utility trade-off across all three settings.

| Data | Attack | FST | | FT | | I-BAU | | FT+SAM | | PL (Ours) | |
|---|---|---|---|---|---|---|---|---|---|---|---|
| | | ASR | C-ACC | ASR | C-ACC | ASR | C-ACC | ASR | C-ACC | ASR | C-ACC |
| *(a) Benign pool contamination $\epsilon_b = 0.1$, $\epsilon_m = 0$* | | | | | | | | | | | |
| CIFAR-10 | Blended | 0.216 | 0.8 | 0.343 | 0.815 | 0.416 | 0.212 | 0.869 | 0.829 | **0.063** | 0.783 |
| | Adaptive Blend | 0.243 | 0.797 | 0.328 | 0.81 | 0.475 | 0.19 | 0.856 | 0.819 | **0.09** | 0.787 |
| | SSBA | 0.066 | 0.792 | 0.057 | 0.808 | 0.113 | 0.191 | 0.217 | 0.81 | **0.019** | 0.776 |
| | BadNet | 0.204 | 0.807 | 0.335 | 0.823 | 0.597 | 0.193 | 0.955 | 0.829 | **0.05** | 0.778 |
| | WaNet | 0.126 | 0.798 | 0.127 | 0.819 | 1 | 0.1 | 0.458 | 0.816 | **0.037** | 0.78 |
| | Mean | 0.171 | 0.799 | 0.238 | 0.815 | 0.52 | 0.177 | 0.671 | 0.821 | **0.052** | 0.781 |
| GTSRB | Blended | 0.3 | 0.881 | 0.437 | 0.903 | **0.013** | 0.17 | 0.927 | 0.917 | 0.173 | 0.875 |
| | Adaptive Blend | 0.393 | 0.872 | 0.502 | 0.893 | **0.001** | 0.193 | 0.95 | 0.914 | 0.212 | 0.858 |
| | SSBA | 0.296 | 0.879 | 0.567 | 0.897 | **0.005** | 0.162 | 0.975 | 0.9 | 0.117 | 0.893 |
| | BadNet | 0.322 | 0.911 | 0.495 | 0.914 | **0.014** | 0.109 | 0.941 | 0.917 | 0.12 | 0.89 |
| | WaNet | 0.107 | 0.894 | 0.161 | 0.909 | 0.864 | 0.018 | 0.651 | 0.905 | **0.061** | 0.877 |
| | Mean | 0.284 | 0.887 | 0.432 | 0.903 | 0.179 | 0.13 | 0.889 | 0.911 | **0.137** | 0.879 |
| Tiny-ImageNet | Blended | 1 | 0.413 | 1 | 0.688 | 1 | 0.005 | 1 | 0.744 | **0** | 0.696 |
| | Adaptive Blend | 1 | 0.413 | 1 | 0.688 | **0** | 0.003 | 1 | 0.744 | **0** | 0.696 |
| | SSBA | 0.999 | 0.41 | 0.999 | 0.719 | 1 | 0.005 | 1 | 0.757 | **0** | 0.715 |
| | BadNet | 0.994 | 0.406 | 0.994 | 0.716 | 0.985 | 0.004 | 0.994 | 0.756 | **0** | 0.734 |
| | WaNet | 0.986 | 0.413 | 0.986 | 0.726 | **0** | 0.005 | 0.068 | 0.746 | **0** | 0.628 |
| | Mean | 0.996 | 0.411 | 0.996 | 0.707 | 0.597 | 0.004 | 0.812 | 0.749 | **0** | 0.694 |
| Grand Mean | | 0.483 | 0.699 | 0.555 | 0.809 | 0.432 | 0.104 | 0.791 | 0.827 | **0.063** | 0.784 |
| *(b) Malicious pool contamination $\epsilon_b = 0$, $\epsilon_m = 0.1$* | | | | | | | | | | | |
| CIFAR-10 | Blended | 0.03 | 0.802 | 0.193 | 0.821 | 0.45 | 0.227 | 0.306 | 0.827 | **0.001** | 0.616 |
| | Adaptive Blend | 0.049 | 0.793 | 0.173 | 0.817 | 0.173 | 0.137 | 0.231 | 0.82 | **0.002** | 0.568 |
| | SSBA | 0.028 | 0.785 | 0.035 | 0.809 | 0.584 | 0.196 | 0.065 | 0.805 | **0.014** | 0.747 |
| | BadNet | 0.081 | 0.803 | 0.376 | 0.822 | 0.176 | 0.251 | 0.805 | 0.829 | **0.004** | 0.63 |
| | WaNet | 0.035 | 0.8 | 0.074 | 0.819 | 0.986 | 0.118 | 0.209 | 0.818 | **0.007** | 0.704 |
| | Mean | 0.045 | 0.797 | 0.17 | 0.818 | 0.474 | 0.186 | 0.323 | 0.82 | **0.006** | 0.653 |
| GTSRB | Blended | 0.001 | 0.903 | 0.017 | 0.917 | **0** | 0.071 | 0.297 | 0.919 | **0** | 0.671 |
| | Adaptive Blend | **0** | 0.891 | 0.276 | 0.904 | **0** | 0.143 | 0.803 | 0.911 | **0** | 0.575 |
| | SSBA | **0** | 0.896 | **0** | 0.905 | **0** | 0.049 | 0.001 | 0.903 | **0** | 0.514 |
| | BadNet | **0** | 0.894 | 0.016 | 0.917 | 0.005 | 0.113 | 0.203 | 0.916 | 0.009 | 0.551 |
| | WaNet | 0.002 | 0.888 | 0.017 | 0.902 | **0** | 0.047 | 0.077 | 0.902 | **0** | 0.765 |
| | Mean | **0.001** | 0.894 | 0.065 | 0.909 | 0.001 | 0.085 | 0.276 | 0.91 | 0.002 | 0.615 |
| Tiny-ImageNet | Blended | **0** | 0.383 | 0.951 | 0.712 | **0** | 0.005 | 0.981 | 0.756 | **0** | 0.56 |
| | Adaptive Blend | **0** | 0.383 | 0.951 | 0.712 | **0** | 0.005 | 0.981 | 0.756 | **0** | 0.56 |
| | SSBA | **0** | 0.384 | 0.979 | 0.714 | **0** | 0.005 | 0.989 | 0.753 | **0** | 0.616 |
| | BadNet | **0** | 0.384 | 0.988 | 0.721 | **0** | 0.005 | 0.992 | 0.758 | **0** | 0.618 |
| | WaNet | **0** | 0.413 | 0.399 | 0.702 | **0** | 0.006 | 0.762 | 0.753 | **0** | 0.282 |
| | Mean | **0** | 0.389 | 0.854 | 0.712 | **0** | 0.005 | 0.941 | 0.755 | **0** | 0.527 |
| Grand Mean | | 0.015 | 0.693 | 0.363 | 0.813 | 0.158 | 0.092 | 0.513 | 0.828 | **0.002** | 0.598 |
| *(c) Both pools contamination $\epsilon_b = 0.1$, $\epsilon_m = 0.1$* | | | | | | | | | | | |
| CIFAR-10 | Blended | 0.312 | 0.794 | 0.442 | 0.819 | 0.695 | 0.163 | 0.83 | 0.828 | **0.103** | 0.644 |
| | Adaptive Blend | 0.329 | 0.792 | 0.44 | 0.812 | 0.496 | 0.153 | 0.851 | 0.816 | **0.118** | 0.652 |
| | SSBA | 0.078 | 0.79 | 0.097 | 0.808 | 0.17 | 0.163 | 0.258 | 0.805 | **0.031** | 0.717 |
| | BadNet | 0.289 | 0.803 | 0.515 | 0.824 | 0.89 | 0.147 | 0.955 | 0.827 | **0.121** | 0.656 |
| | WaNet | 0.179 | 0.796 | 0.197 | 0.82 | 0.999 | 0.102 | 0.515 | 0.811 | **0.06** | 0.734 |
| | Mean | 0.237 | 0.795 | 0.338 | 0.817 | 0.65 | 0.146 | 0.682 | 0.817 | **0.087** | 0.681 |
| GTSRB | Blended | 0.442 | 0.908 | 0.607 | 0.918 | **0.006** | 0.155 | 0.937 | 0.923 | 0.023 | 0.663 |
| | Adaptive Blend | 0.474 | 0.897 | 0.617 | 0.905 | 0.011 | 0.138 | 0.935 | 0.914 | **0.001** | 0.488 |
| | SSBA | 0.283 | 0.882 | 0.649 | 0.9 | 0.017 | 0.227 | 0.974 | 0.905 | **0** | 0.517 |
| | BadNet | 0.684 | 0.918 | 0.793 | 0.923 | 0.152 | 0.164 | 0.948 | 0.924 | **0.002** | 0.614 |
| | WaNet | 0.168 | 0.891 | 0.246 | 0.906 | 0.736 | 0.012 | 0.659 | 0.903 | **0.045** | 0.741 |
| | Mean | 0.41 | 0.899 | 0.582 | 0.91 | 0.184 | 0.139 | 0.891 | 0.914 | **0.014** | 0.605 |
| Tiny-ImageNet | Blended | 1 | 0.403 | 1 | 0.719 | 0.991 | 0.026 | 1 | 0.756 | **0** | 0.546 |
| | Adaptive Blend | 1 | 0.403 | 1 | 0.719 | 0.991 | 0.026 | 1 | 0.756 | **0** | 0.546 |
| | SSBA | 0.999 | 0.415 | 1 | 0.715 | 1 | 0.005 | 1 | 0.757 | **0** | 0.579 |
| | BadNet | 0.994 | 0.397 | 0.995 | 0.725 | 0.976 | 0.005 | 0.994 | 0.761 | **0.001** | 0.559 |
| | WaNet | 0.993 | 0.406 | 0.992 | 0.713 | **0** | 0.005 | 0.1 | 0.748 | **0** | 0.563 |
| | Mean | 0.997 | 0.405 | 0.997 | 0.718 | 0.792 | 0.013 | 0.819 | 0.756 | **0** | 0.559 |
| Grand Mean | | 0.548 | 0.7 | 0.639 | 0.815 | 0.542 | 0.099 | 0.797 | 0.829 | **0.034** | 0.615 |

is not trained on the target attack, often matching or beating the model fine-tuned directly on that attack. PL generalizes to unseen backdoor triggers in such scenarios. Under malicious and combined contamination, some degradation occurs, but there are no catastrophic failures. These results demonstrate that PL retains robustness even when the attack type at test time differs from the one used for fine-tuning, and that the method degrades gracefully as contamination increases.

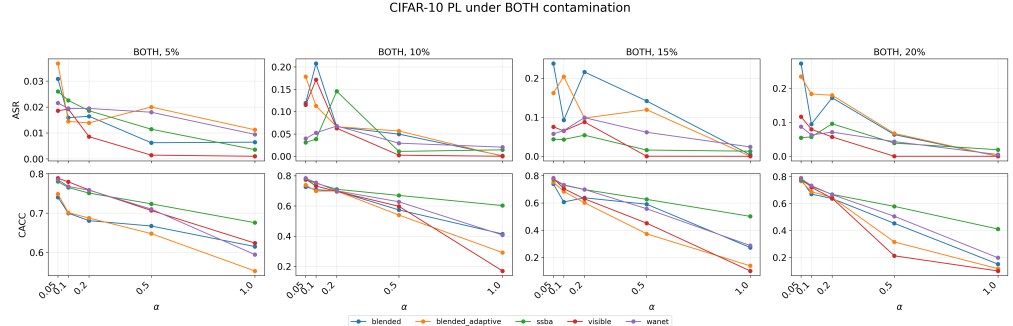

Figure 6: **PL under combined contamination (FN+FP).**

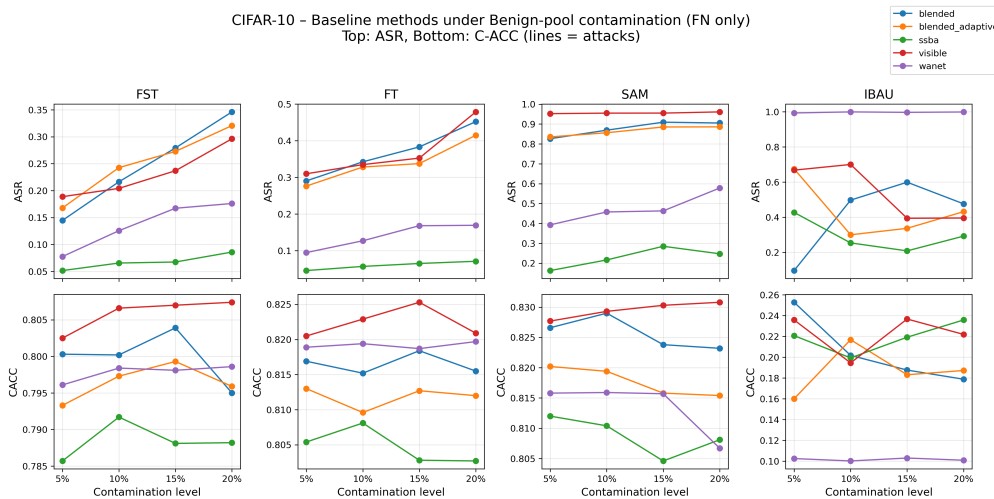

Figure 7: **Baseline under benign contamination (FN only).**

## F   USE OF LLMS

We used a large language model (LLM) to aid and polish the writing of this paper. All ideas, methodology, experiments, and analyses are original contributions of the authors.

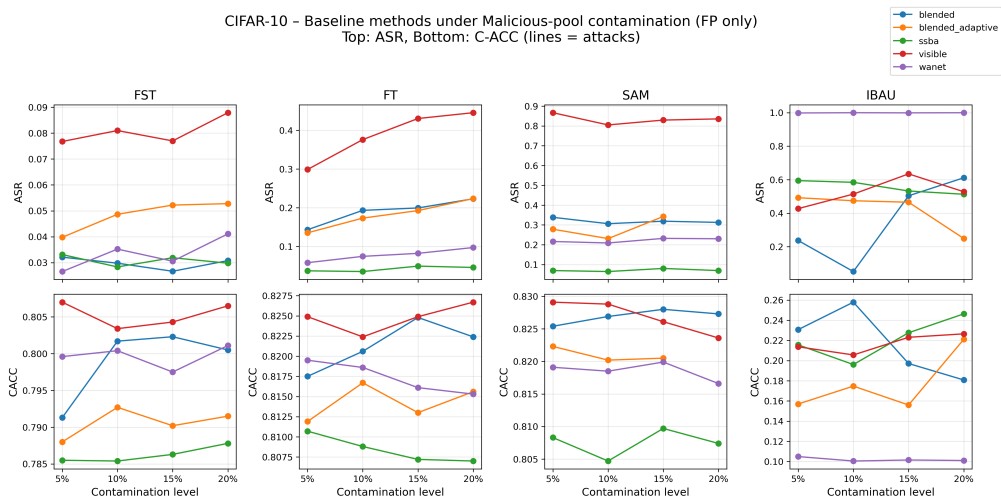

Figure 8: **Baseline under malicious contamination (FP only).**

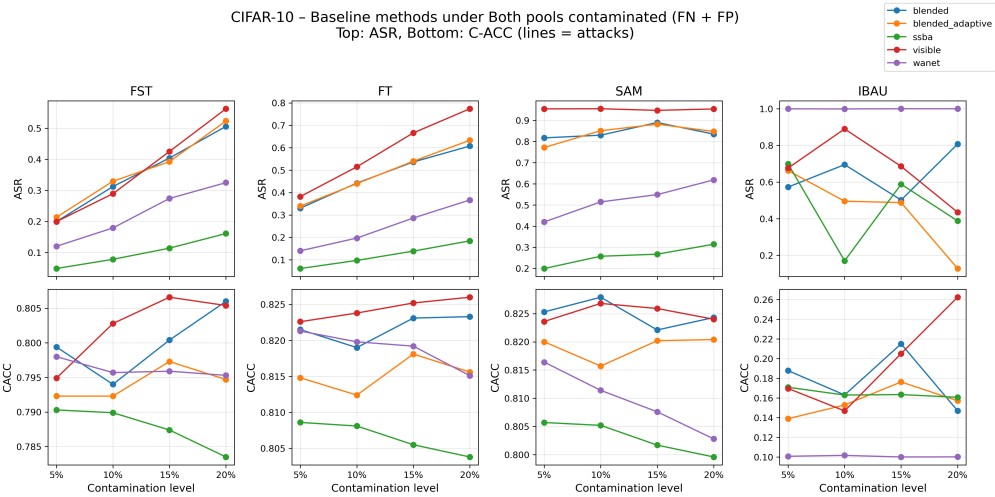

Figure 9: **Baseline under combined contamination (FN+FP).**

Table 4: PL ($\alpha = 0.2$) performance under clean partition. Each cell reports C-ACC / ASR.

|                   | blended     | blended_adaptive | ssba        | visible     | wanet       |
|-------------------|-------------|------------------|-------------|-------------|-------------|
| blended           | 0.795/0.006 | 0.795/0.006      | 0.795/0.007 | 0.795/0.009 | 0.795/0.010 |
| blended_adaptive  | 0.785/0.012 | 0.785/0.012      | 0.785/0.007 | 0.785/0.008 | 0.785/0.010 |
| ssba              | 0.774/0.007 | 0.774/0.007      | 0.774/0.013 | 0.774/0.008 | 0.774/0.007 |
| visible           | 0.774/0.001 | 0.774/0.001      | 0.774/0.002 | 0.774/0.021 | 0.774/0.002 |
| wanet             | 0.771/0.001 | 0.771/0.001      | 0.771/0.003 | 0.771/0.005 | 0.771/0.008 |

Table 5: PL ($\alpha = 0.2$, Train_prop = 5%) under different contamination regimes. Each cell shows C-ACC / ASR.

*(a) Benign pool contamination: $\epsilon_b = 0.1$, $\epsilon_m = 0$ (False Negatives only)*

|                  | blended     | blended_adaptive | ssba        | visible     | wanet       |
| ---------------- | ----------- | ---------------- | ----------- | ----------- | ----------- |
| blended          | 0.783/0.063 | 0.783/0.063      | 0.783/0.005 | 0.783/0.006 | 0.783/0.005 |
| blended_adaptive | 0.787/0.090 | 0.787/0.090      | 0.787/0.009 | 0.787/0.012 | 0.787/0.012 |
| ssba             | 0.776/0.006 | 0.776/0.006      | 0.776/0.019 | 0.776/0.006 | 0.776/0.006 |
| visible          | 0.778/0.001 | 0.778/0.001      | 0.778/0.001 | 0.778/0.050 | 0.778/0.001 |
| wanet            | 0.780/0.003 | 0.780/0.003      | 0.780/0.004 | 0.780/0.006 | 0.780/0.037 |

*(b) Malicious pool contamination: $\epsilon_b = 0$, $\epsilon_m = 0.1$ (False Positives only)*

|                  | blended     | blended_adaptive | ssba        | visible     | wanet       |
| ---------------- | ----------- | ---------------- | ----------- | ----------- | ----------- |
| blended          | 0.616/0.001 | 0.616/0.001      | 0.616/0.183 | 0.616/0.270 | 0.616/0.334 |
| blended_adaptive | 0.568/0.002 | 0.568/0.002      | 0.568/0.276 | 0.568/0.357 | 0.568/0.425 |
| ssba             | 0.747/0.020 | 0.747/0.020      | 0.747/0.014 | 0.747/0.039 | 0.747/0.037 |
| visible          | 0.629/0.256 | 0.629/0.256      | 0.629/0.219 | 0.629/0.004 | 0.629/0.216 |
| wanet            | 0.704/0.077 | 0.704/0.077      | 0.704/0.048 | 0.704/0.058 | 0.704/0.007 |

*(c) Both pools contamination: $\epsilon_b = 0.1$, $\epsilon_m = 0.1$ (FN + FP)*

|                  | blended     | blended_adaptive | ssba        | visible     | wanet       |
| ---------------- | ----------- | ---------------- | ----------- | ----------- | ----------- |
| blended          | 0.644/0.103 | 0.648/0.110      | 0.672/0.151 | 0.673/0.585 | 0.640/0.557 |
| blended_adaptive | 0.648/0.110 | 0.652/0.118      | 0.676/0.134 | 0.677/0.561 | 0.644/0.526 |
| ssba             | 0.722/0.461 | 0.714/0.474      | 0.717/0.031 | 0.729/0.401 | 0.718/0.317 |
| visible          | 0.663/0.570 | 0.668/0.556      | 0.676/0.199 | 0.656/0.121 | 0.671/0.327 |
| wanet            | 0.716/0.493 | 0.704/0.499      | 0.717/0.101 | 0.732/0.480 | 0.734/0.060 |

