# OpenReview forum: "Partition-Losses Fine-Tuning: Contamination-Robust Backdoor Unlearning"
_ICLR.cc/2026/Conference — Submitted to ICLR 2026_

### Official Review · Reviewer_ZcSL · 2025-10-26

**Soundness:** 2
**Presentation:** 3
**Contribution:** 2
**Rating:** 2
**Confidence:** 5

**Summary:**

The authors propose Partition-Losses Fine-Tuning (PL), a simple, architecture- and domain-agnostic loss modification that leverages both clean and flagged malicious samples in DNN to mitigate backdoor threats. PL jointly minimizes benign loss and maximizes target-class loss, explicitly pushing the model away from the implanted trigger-to-target association.

**Strengths:**

The paper is well-structured, clearly organized and presented.

**Weaknesses:**

1.Unrealistic Threat Model: The paper is partially motivated on the realism of the assumption that tuning on clean data to eliminate backdoors. However, while the PL method reduces the required number of clean samples by half and allows for contamination, it still necessitates backdoor-labeled malicious samples. How is this assumption ensured in practice?

2.Missing Technical Details -- Collection of $D_m$: The method mentioned for "flagging malicious inputs with the target class involves real-time monitoring and incident-response pipelines". What specifically does this entail? Additionally, I did not see theoretical validation or ablation studies regarding the scales of $D_m$ and $D_b$, which I believe are crucial for the implementation of PL and partly determine the sophistication of the objective function.

3.Missing Technical Details -- |$D_m$|=0: Although line 171 mentions, “Note that PL reduces exactly to clean-only fine-tuning when no flagged triggered samples are available,” this method is not discussed further. What does this method entail, and how's its performance?

4.Theoretical Analysis: There is a lack of discussion regarding $D_m$ and $D_b$, along with upper and lower bound validations.

5.Evaluation of DNN Tasks: For text and tabular data, as well as classification and regression tasks, the methods of implanting triggers and training backdoors differ. I would like to see implementation modifications of PL across these DNN tasks and the corresponding results.

6.Choice of Evaluation Settings: The methods used in the attack settings lacks the reflection of SOTA approaches, and there is insufficient detailed description of the implementations of Adaptive Blend and SSBA methods.

7.Insufficient Discussion on the Impact of Different $\epsilon$ on PL Performance: The discussion is limited to cases where $\epsilon_b$ and $\epsilon_m$ are both 0.1. However, according to the proof section, “when $\epsilon_m$ < 1/2 , such an alpha always exists.” The boundary conditions and strategies for selecting alpha under different $\epsilon$ values are not discussed. Additionally, why is it claimed in line 431 that both $\epsilon$ = 0.1 is “realistic contamination”? What is the basis for this statement?

8.Outdated Related Work: In the related work section, the citations regarding fine-tuning range from 2018 to 2023 and do not include recent SOTA contributions.

**Questions:**

1.What do the real-time monitoring and incident-response pipelines for flagging malicious inputs with the target class specifically entail? What are their success and false positive rates?

2.Can the authors further discuss the scale or ratio of $D_m$ and $D_b$, including theoretical derivation and empirical evidence?

3.What is the methodology of the deployed attack methods in the evaluation setting? The baseline defense methods perform poorly in this context; under what settings were the experiments conducted in the papers they are proposed?

---

> ### Author Response · Authors · 2025-11-20
>
> We are grateful to the reviewer for the detailed assessment and suggestions, which helped us improve the clarity and presentation of our contributions!
>
> > **Weakness 1**: Unrealistic Threat Model: The paper is partially motivated on the realism of the assumption that tuning on clean data to eliminate backdoors. However, while the PL method reduces the required number of clean samples by half and allows for contamination, it still necessitates backdoor-labeled malicious samples. How is this assumption ensured in practice?
>
> Since Neural Cleanse and TABOR, many follow-up backdoor detection methods (e.g., B3D/B3D-SS[1], CBDSCM [2], BAN[3], BBCaL[4], PSBD[5], UFID[6]) have reported detection accuracies typically above 95% on commonly used benchmark datasets in the computer vision field. The ever-improving performance of high-accuracy detectors makes it realistic for us defenders to obtain high-confidence backdoor samples along with the attack label. Our contribution is complementary and targets the realistic incident-response phase. Given such a small set of suspected malicious samples, PL is a principled, optimization-based fine-tuning method that remains effective even with contaminated or limited amounts of data.
>
> **[1]** Dong, Yinpeng et al. *Black-box detection of backdoor attacks with limited information and data.* ICCV, 2021.
> **[2]** Xian, Xun et al. *A unified detection framework for inference-stage backdoor defenses.* NeurIPS, 2023.
> **[3]** Xu, Xiaoyun et al. *BAN: Detecting backdoors activated by adversarial neuron noise.* NeurIPS, 2024.
> **[4]** Hu, Mengxuan et al. *BBCaL: Black-box Backdoor Detection under the Causality Lens.* TMLR, 2024.
> **[5]** Li, Wei et al. *PSBD: Prediction shift uncertainty unlocks backdoor detection.* CVPR, 2025.
> **[6]** Guan, Zihan et al. *UFID: A Unified Framework for Black-box Input-level Backdoor Detection.* AAAI, 2025.
>
> > **Weakness 2**: Missing Technical Details -- Collection of D_m: The method mentioned for "flagging malicious inputs with the target class involves real-time monitoring and incident-response pipelines". What specifically does this entail? Additionally, I did not see theoretical validation or ablation studies regarding the scales of D_m and
> D_b, which I believe are crucial for the implementation of PL and partly determine the sophistication of the objective function.
>
> Our statement on "flagging malicious inputs with the target class involves real-time monitoring and incident-response pipelines" refers to the recognition of existing, well-established backdoor detection mechanisms. Numerous successful detectors can provide >95% accuracy in identifying suspicious input [1,2,3,4,5,6]. These high-accuracy detectors could be routinely used on the commonly used benchmark dataset in the computer vision field. Our assumption of having access to detected suspicious samples is therefore consistent with the progress in the backdoor field. Additionally, PL does not require perfect detection; a coarse partition of the tuning set is sufficient to provide good performance (low ASR and high C-ACC).
>
> We thank the reviewer for highlighting the "scales of D_m and D_b". PL is explicitly designed to operate under an imperfect and imbalanced tuning pool. In the main paper (Table 2) and supplementary (Table 3), we evaluate PL at different data availability under: 1) benign contamination: 10% are malicious samples mislabeled as benign in D_B; 2) malicious contamination: 10% are benign samples mislabeled as malicious in D_M; 3) both: 10% contamination in each pool. Across all settings, PL consistently achieves the lowest attack success rate (ASR) while maintaining competitive clean accuracy.
>
> To further address the review's questions on scales, we added new experiments where contamination varies from 5%, 10%, 15%, to 20% under malicious contamination (FP-only), benign contamination (FN-only), and combined contamination (FP+FN). The results show (Figures 4–9, Appendix D): PL performance degrades smoothly, not catastrophically; Moderate $\alpha$ values (0.1–0.5) remain robust across all scales. These findings confirm that PL is resilient to both imbalance and noise in the pool partition.

---

> ### Author Response · Authors · 2025-11-20
>
> > **Weakness 3**: Missing Technical Details -- |D_m|=0: Although line 171 mentions, “Note that PL reduces exactly to clean-only fine-tuning when no flagged triggered samples are available,” this method is not discussed further. What does this method entail, and how's its performance?
>
> Line 171 refers to the limiting case where the PL regularization term disappears, because there are no flagged malicious samples available to build this contrastive penalty. Thus, the PL objective reduces exactly to ordinary clean-only fine-tuning.
>
> The behavior is consistent with our empirical findings. In Figure 2 (main paper) and in the new contamination-sweep experiments (Appendix D). When the penalty weight $\alpha$ is very small (e.g., 0.05 or 0.1), the regularization term becomes negligible, and the model behaves similarly to the clean-only fine-tuning, where the backdoor behaviors are preserved. It achieves relatively higher clean accuracy but also very high ASR, since no repulsive gradient is present to suppress the backdoor.
>
> Thus, |D_M|=0 corresponds to the extreme where $\alpha$ becomes zero, and our experiments captured this regime.
>
> > **Weakness 4**: Theoretical Analysis: There is a lack of discussion regarding D_m and D_b, along with upper and lower bound validations.
>
> In Section 3, Theorem 1, we explicitly characterize how the contamination ratio interacts with both the objective and the guarantees. We provide (i) an upper bound on tolerable contamination ($\epsilon_m<0.5$) and (ii) a constructive lower bound on $\alpha$ that guarantees monotone ASR decrease.
>
> Part (1) of Theorem 1 shows that as long as the malicious pool is not overly contaminated, e.g., $\epsilon_m<1/2$, the expected gradient step on L_PL reduces the surrogate attack success rate more than the clean-only fine-tuning, for any regularization $\alpha >0$. This provides an explicit upper bound on the admissible contamination of  D_m. As long as a majority of the malicious pool consists of truly triggered samples, PL is guaranteed to suppress the backdoor more aggressively than a standard fine-tuning.
> Part (2) gives a sufficient lower bound on the trade-off parameter $\alpha$. Under the given condition, the PL objective $L_PL$ itself strictly decreases the surrogate ASR at each optimization step. When $\epsilon_m<1/2$, such $\alpha$ always exists.
>
> > **Weakness 5**: Evaluation of DNN Tasks: For text and tabular data, as well as classification and regression tasks, the methods of implanting triggers and training backdoors differ. I would like to see implementation modifications of PL across these DNN tasks and the corresponding results.
>
> We agree that extending PL beyond image classification is an important future direction. We restrict our empirical evaluation to the vision setting for two reasons:
> - 1) The backdoor-defense baselines we compare against (FST, FT, SAM, I-BAU) are developed and reported on image classification datasets. To ensure a fair and controlled comparison, our experiments follow the standard vision evaluation protocol.
> - 2) Conceptually, PL is modality-agnostic. The PL objective only assumes a supervised classifier with a probabilistic output $f_\theta(x)$, two tuning sets. The same regularization can be applied to the text and tabular backdoors by pairing it with an appropriate backbone model.
>
> It is a promising next step to study how the trigger mechanism differs across modalities, and we appreciate the reviewer highlighting this direction.

---

> ### Author Response · Authors · 2025-11-20
>
> >**Weakness 6**: Choice of Evaluation Settings: The methods used in the attack settings lacks the reflection of SOTA approaches, and there is insufficient detailed description of the implementations of Adaptive Blend and SSBA methods.
>
> We thank the reviewer for pointing out the technical details. We have elaborated the descriptions of Adaptive Blend and SSBA in the revised Appendix (Section Experimental Setup, Attack Settings)
>
> SSBA[1] is a dynamic invisible-trigger attack. We implement the complete SSBA encoder–decoder pipeline following Li et al.[1]: we train the SSBA steganographic trigger, where a learnable encoder injects a binary fingerprint into each input image, and a decoder is jointly trained to recover this fingerprint at a poisoning rate of 0.15.
>
> Adaptive Blend[2] is an enhanced variant of the classical blended attack. It consists of input-dependent partial blending and probabilistic label flipping. During poisoning, the image is divided into a grid of patches; only half of the patches are randomly selected for blending with the trigger at a poisoning rate of 0.15. The remaining patches stay unchanged, creating a spatially sparse and input-dependent backdoor pattern.
>
> These clarifications have been added to the appendix to ensure full reproducibility.
>
> **[1]** Li, Yuezun, Yiming Li, Baoyuan Wu, Longkang Li, Ran He, and Siwei Lyu. Invisible Backdoor Attack with Sample-Specific Triggers. ICCV, 2021.
> **[2]** Qi, Xiangyu; Xie, Tinghao; Li, Yiming; Mahloujifar, Saeed; and Mittal, Prateek. “Revisiting the Assumption of Latent Separability for Backdoor Defenses.” ICLR 2023.
>
> > **Weakness 7**: Insufficient Discussion on the Impact of Different on PL Performance: The discussion is limited to cases where D_m and D_b are both 0.1. However, according to the proof section, “when e_m < 1/2 , such an alpha always exists.” The boundary conditions and strategies for selecting alpha under different values are not discussed. Additionally, why is it claimed in line 431 that both e = 0.1 is “realistic contamination”? What is the basis for this statement?
>
> In the main text, we evaluated PL under 10% contamination. This reflects realistic conditions after applying modern backdoor detection systems. Recent detectors [3,4,5,6,7,8] routinely achieve >95% detection accuracy and high F1 scores on CIFAR-10, GTSRB, and ImageNet benchmarks.
>
> The theoretical analysis of PL does not assume $\epsilon_m = 0.1$. Theorem 1 shows that as long as more than half of the contaminated pool are truly trigger samples, there always exists a trade-off parameter $\alpha$ that guarantees a strict reduction in surrogate attack success.
>
> We additionally include new experiments on contamination from 5% to 20% under benign, malicious, and combined contamination. Across all settings, PL only degrades slightly at higher contamination rates rather than catastrophically. The dominance over baselines persists (Figures 4–6, Appendix D).
>
> **[3]** Dong, Yinpeng et al. Black-box detection of backdoor attacks with limited information and data. ICCV, 2021.
> **[4]** Xian, Xun et al. A unified detection framework for inference-stage backdoor defenses. NeurIPS, 2023.
> **[5]** Xu, Xiaoyun et al. BAN: Detecting backdoors activated by adversarial neuron noise. NeurIPS, 2024.
> **[6]** Hu, Mengxuan et al. BBCaL: Black-box Backdoor Detection under the Causality Lens. TMLR, 2024.
> **[7]** Li, Wei et al. PSBD: Prediction shift uncertainty unlocks backdoor detection. CVPR, 2025.
> **[8]** Guan, Zihan et al. UFID: A Unified Framework for Black-box Input-level Backdoor Detection. AAAI, 2025.
>
> > **Weakness 8**: Outdated Related Work: In the related work section, the citations regarding fine-tuning range from 2018 to 2023 and do not include recent SOTA contributions.
>
> We thank the reviewer for this comment. We are unaware of any recent work that has provided a stronger finetuning backdoor defense than Feature Shift Tuning (FST). Relaxing the assumption of clean data for fine-tuning defenses is a novel contribution that considerably broadens the scope of such defenses. If there are particular papers the reviewer believes are essential for inclusion, we would be happy to incorporate them.

---

> ### Author Response · Authors · 2025-11-20
>
> > **Question 1**: What do the real-time monitoring and incident-response pipelines for flagging malicious inputs with the target class specifically entail? What are their success and false positive rates?
>
> We thank the reviewer for raising this point. We have updated the Introduction with a more detailed elaboration. In industry and large-scale ML deployments, real-time monitoring and incident-response pipelines refer to standard practices to detect abnormal model behavior during inference. In the context of backdoor detection, these pipelines rely on existing state-of-the-art (SOTA) inference-time backdoor detectors that automatically flag suspicious inputs. Modern backdoor detectors are highly accurate. These detectors achieve very high recall (typically >95%) with low false-positive rates. Using their flagged samples to form D_M is considered realistic and aligns with the assumptions of prior work.
>
> > **Question 3**: What is the methodology of the deployed attack methods in the evaluation setting? The baseline defense methods perform poorly in this context; under what settings were the experiments conducted in the papers they are proposed?
>
> We use the standard, widely adopted implementations of five representative backdoor attacks:
> - Blended [1]: linear trigger blending on input pixels,
> - Blended-adaptive [2], an enhanced Blended attack that applies spatially partitioned partial blending and asymmetric opacity,
> - SSBA [3]: sample-specific trigger generation,
> - Visible [4]: fixed patch-based trigger,
> - WaNet [5]: smooth, image-dependent geometric warping.
>
> The four baseline defenses(FST [6], FT[6], I-BAU[7], FT+SAM[8]) that we compare against were not designed for contaminated fine-tuning. In their original papers, all baseline defenses assume perfect access to a clean fine-tuning dataset. Under those idealized settings, the baselines perform strongly. They match or exceed PL under perfect partition in Tables 2 and 3 (Main text).
>
> Our work targets the practical scenario that the baseline papers do not address, where the fine-tuning set is contaminated. Thus, their weaker performance in our setting is not surprising. These defenses were simply not designed for this reality. In contrast, PL was specifically designed for this contaminated-partition scenario. And the extensive empirical and theoretical research has demonstrated its efficacy.
>
> **[1]** Chen, Xinyun, et al. "Targeted backdoor attacks on deep learning systems." 2017.
> **[2]** Qi, Xiangyu, et al. "Revisiting the assumption of latent separability for backdoor defenses." ICLR, 2023.
> **[3]** Li, Yuezun, et al. "Invisible backdoor attack with sample-specific triggers." CVPR, 2021.
> **[4]** Gu, Tianyu, et al. "BadNets: Evaluating backdoor attacks on deep neural networks." 2017.
> **[5]** Nguyen, Anh, and Anh Tran. "WaNet: Imperceptible warping-based backdoor attack." ICLR, 2021.
> **[6]** Min, Rui, Zeyu Qin, Li Shen, and Minhao Cheng. "Towards stable backdoor purification through feature shift tuning." NeurIPS, 2023.
> **[7]** Zhao, Zhaoyuan, et al. "I-BAU: Indistinguishable backdoor attack unlearning." NeurIPS, 2022.
> **[8]** Zhu, Mingli, Shaokui Wei, Li Shen, Yanbo Fan, and Baoyuan Wu. "Enhancing fine-tuning based backdoor defense with sharpness-aware minimization." ICCV, 2023.

---

> > ### Comment · Reviewer_ZcSL · 2025-11-28
> >
> > I believe the response addresses my primary concerns. The scientific contribution and presentation are significantly improved.

---

### Official Review · Reviewer_UZFP · 2025-10-31

**Soundness:** 3
**Presentation:** 2
**Contribution:** 1
**Rating:** 4
**Confidence:** 2

**Summary:**

This paper proposes Partition-Losses Fine-Tuning (PL), a simple and robust post-training defense against poisoning-based backdoor attacks. Unlike traditional fine-tuning methods that assume completely clean data, PL leverages both benign and flagged malicious samples by minimizing the loss on clean data while maximizing the loss on triggered samples. This dual-objective design explicitly unlearns the trigger-to-target association, making the model more resistant to contamination. Experiments across multiple datasets and attack types show that PL consistently achieves near-zero attack success rates with competitive clean accuracy, even when the fine-tuning data is partially contaminated. The method is lightweight, architecture-agnostic, and practical for real-world deployment in security-critical applications.

**Strengths:**

The main strength of Partition-Losses Fine-Tuning (PL) lies in its robustness, simplicity, and effectiveness. It performs well even with contaminated fine-tuning data, requires no access to the original training set, and is easy to implement. PL achieves near-zero attack success rates while maintaining high clean accuracy across different models and attacks, making it a practical and generalizable defense for real-world applications.

**Weaknesses:**

The main weakness of Partition-Losses Fine-Tuning (PL) is its dependence on prior knowledge of the backdoor trigger and target label, as well as its vulnerability to data partitioning errors.

PL assumes the defender can identify or flag a small set of triggered samples and their associated target class to form the malicious pool \( D_m \).
However, in realistic scenarios, this assumption is often unrealistic — triggers and target labels are typically unknown or mislabeled.
When the partition between the benign pool \( D_b \) and malicious pool \( D_m \) is inaccurate (e.g., due to high false-positive or false-negative rates), the joint optimization objective can become misleading, leading to:

1. Incomplete backdoor removal — residual trigger-target associations may persist.
2. Clean accuracy degradation — benign knowledge may be unintentionally unlearned.

Additionally, the trade-off parameter \( \alpha \), which balances clean accuracy and backdoor forgetting, is static and sensitive:
1. A large \( \alpha \) can severely harm benign accuracy.
2. A small \( \alpha \) may fail to erase the backdoor.

Compared with recent adaptive or self-supervised defenses (e.g., exposure-based or gradient inversion methods), PL lacks dynamic control and trigger-agnostic robustness.

**Questions:**

1. **Trigger Identification**
   - How can PL be applied when the defender has *no access* to labeled triggered samples or the target class?
   - Can the method be extended to *automatically detect* or *estimate* potential triggers?

2. **Partition Robustness**
   - How sensitive is PL to mislabeling or contamination in the benign/malicious pools?
   - What happens when the contamination rate exceeds the tested threshold (e.g., >10%)?

3. **Parameter Sensitivity**
   - How should the trade-off coefficient \( \alpha \) be chosen or adapted during training?
   - Can an *adaptive α-scheduler* improve stability between backdoor forgetting and clean accuracy?

4. **Generalization to Unknown Attacks**
   - Does PL remain effective against *adaptive* or *unknown trigger* attacks that were not seen during fine-tuning?
   - How does it perform on *semantic* or *dynamic* triggers instead of static patterns?

5. **Comparison with Modern Defenses**
   - How does PL perform compared to *exposure-based*, *representation-level unlearning*, or *causal filtering* defenses introduced in 2024–2025?
   - Can PL be combined with these methods to form a hybrid, more resilient defense?

---

> ### Author Response · Authors · 2025-11-20
>
> We thank Reviewer for the detailed feedback and for highlighting important points regarding the experimental setup!
>
> > Weakness: The main weakness of Partition-Losses Fine-Tuning (PL) is its dependence on prior knowledge of the backdoor trigger and target label, as well as its vulnerability to data partitioning errors.
>
> We actually do not assume knowledge of the trigger or target label. Instead, we presume access only to samples that have been partitioned, with fewer than half of the malicious samples in the mostly-clean partition. This partitioning could have been created from any (possibly multiple) backdoor triggers or target labels and by any detection model. Hence, our proposal is applicable to a wide range of backdoor defense settings.
>
> > PL assumes the defender can identify or flag a small set of triggered samples and their associated target class to form the malicious pool (D_m). However, in realistic scenarios, this assumption is often unrealistic — triggers and target labels are typically unknown or mislabeled.
>
> We agree that triggers and target labels can be mislabeled. The primary motivation behind our method is that we are the first finetuning-based backdoor defense to relax the requirement for access to clean data. Instead, we only need a detection model that is better than random chance, which is a huge relaxation of this field’s typical assumption of guaranteed clean data.
>
> > When the partition between the benign pool ( D_b ) and malicious pool ( D_m ) is inaccurate (e.g., due to high false-positive or false-negative rates), the joint optimization objective can become misleading, leading to:
> > 1. Incomplete backdoor removal — residual trigger-target associations may persist.
> > 2. Clean accuracy degradation — benign knowledge may be unintentionally unlearned.
>
> Both of the above issues are well-known concerns for all backdoor defense methods and are not special to our joint optimization objective. However, a user can evaluate the clean accuracy degradation and decide how much performance loss they are comfortable with when deciding whether to use (or not) a finetuned model. Most importantly, regarding incomplete backdoor removal, we have extensive experimental results (Tables 2–3 (main §4.2) and Figures 4–9 (Appendix D)) demonstrating that our proposal provides stronger backdoor removal than other finetuning-based backdoor defenses, with or without contamination in data partitioning.
>
> > Additionally, the trade-off parameter \( \alpha \), which balances clean accuracy and backdoor forgetting, is static and sensitive:
> >
> > - A large \( \alpha \) can severely harm benign accuracy.
> > - A small \( \alpha \) may fail to erase the backdoor.
> >
> > Compared with recent adaptive or self-supervised defenses (e.g., exposure-based or gradient-inversion methods), PL lacks dynamic control and trigger-agnostic robustness.
>
>
> Having a static hyperparameter is not necessarily a disadvantage and can often be more interpretable. With our current setup, we can interpret $\alpha$ as the degree of emphasis we place on unlearning behavior from the “predicted malicious” partition, which can be directly influenced by one's beliefs about the prevalence and severity of possible attacks for a given setting. Actually, our trade-off parameter \alpha  could be made dynamic. For example, we could use soft partitioning (based on the probability that a sample is “malicious”) to weight the regularization gradient rather than determining which partition a sample belongs to. The reason such a dynamic hyperparameter is outside the scope of this work is that we focus on how to unlearn backdoor behavior using data and partitioning, making our proposal applicable to most detection frameworks. Future work could and should examine how targeted detection can improve our unlearning.

---

> ### Author Response · Authors · 2025-11-20
>
> > **Question 1: Trigger Identification**
> >
> > - How can PL be applied when the defender has no access to labeled triggered samples or the target class?
> > - Can the method be extended to automatically detect or estimate potential triggers?
>
> Our method, PL, does not require access to the attack's original triggered samples or the attack class. Instead, PL operates in the post-detection setting, where defenders surface a small set of suspicious inputs and the likely target class. Since Neural Cleanse and TABOR, backdoor detection at the inference stage has been extensively studied [1,2,3,4,5,6]. Those proposed detectors can routinely achieve >95% detection accuracy on commonly used benchmark datasets in the computer vision field (CIFAR10, GTSRB, ImageNet). The empirical success makes it realistic for us, as defenders, to obtain a small set of high-confidence backdoored samples along with the suspected target label.
>
> **[1]** Dong, Yinpeng et al. Black-box detection of backdoor attacks with limited information and data. ICCV, 2021.
> **[2]** Xian, Xun et al. A unified detection framework for inference-stage backdoor defenses. NeurIPS, 2023.
> **[3]** Xu, Xiaoyun et al. BAN: Detecting backdoors activated by adversarial neuron noise. NeurIPS, 2024.
> **[4]** Hu, Mengxuan et al. BBCaL: Black-box Backdoor Detection under the Causality Lens. TMLR, 2024.
> **[5]** Li, Wei et al. PSBD: Prediction shift uncertainty unlocks backdoor detection. CVPR, 2025.
> **[6]** Guan, Zihan et al. UFID: A Unified Framework for Black-box Input-level Backdoor Detection. AAAI, 2025.
>
> > **Question 2: Partition Robustness**
> >
> > - How sensitive is PL to mislabeling or contamination in the benign/malicious pools?
> > - What happens when the contamination rate exceeds the tested threshold (e.g., >10%)?
>
> Across FP, FN, and FP+FN settings, moderate $\alpha$ values ([0.1, 0.5]) consistently yield low ASR with only a modest drop in C-ACC. Very small $\alpha (0.05)$ can leave residual backdoor signatures (higher ASR), whereas very large $\alpha(1.0)$ over-regularizes and harms C-ACC. These trends align with the pattern observed in Figure 2 of the main paper: ASR decreases with increasing $\alpha$, while C-ACC collapses when $\alpha$ is pushed to extreme values.
>
> In Table 2, we provide a comprehensive comparison between PL and four baseline defenses under three partition regimes of the fine-tuning set (perfect partition, benign contamination, and malicious contamination). Across all settings, PL consistently achieved the lowest ASR while maintaining competitively high clean accuracy (C-ACC).
>
> To further study cases with higher contamination levels, we ran additional experiments on the CIFAR-10 (Figures 4–9, Appendix D). We varied the contamination rate in the two pools at {5%, 10%, 15%, 20%}. We considered three contamination types: false positives only (FP), false negatives only (FN), and combined FP+FN. For each setting, we evaluated the performance of PL with $\alpha \in$ {0.05, 0.1, 0.2, 0.5, 1.0}.
>
> Overall, we do not observe catastrophic failure as contamination rates increase from $5\%$ to $20\%$. PL's C-ACC and ASR change smoothly. Even at a $20\%$ contamination rate in both benign and malicious tuning pools, PL could still maintain high C-ACC and low ASR with a reasonable choice of $\alpha$.
>
> At lower or higher contamination cases, across FP, FN, and FP+FN settings, moderate $\alpha$ values ($[0.1,0.5]$) consistently produce low ASR with only a modest drop in C-ACC. Very small $\alpha (0.05$) sometimes leaves residual backdoor effects (higher ASR), whereas very large $\alpha (1.0$) over-regularizes and hurts C-ACC. These observations show a consistent pattern, as shown in Figure 2 in the main paper. The ASR decreases with alpha, while C-ACC collapses when alpha is pushed to extreme values.

---

> ### Author Response · Authors · 2025-11-20
>
> > **Question 3: Parameter Sensitivity**
> >
> > - How should the trade-off coefficient ( \alpha ) be chosen or adapted during training?
> > - Can an adaptive α-scheduler improve stability between backdoor forgetting and clean accuracy?
>
> In principle, an adaptive scheme could automate the trade-off. We consider it as an orthogonal extension for further improvement, not a core limitation of PL. Several standard strategies could be plugged in.
>
> Validation-driven adaptation can periodically evaluate on a small validation set and adjust the $\alpha$ up or down based on whether the performance is above or below the target ASR or C-ACC thresholds. Constraint-based Lagrangian update treats ASR as a constraint and updates alpha as a dual variable, so $\alpha$ decreases when ASR is already low, but C-ACC starts to drop and then increase when ASR is too high [1]. Warm-restart schedule can similarly modulate $\alpha$ to improve optimization stability [2].
>
> Exploring such adaptive schedules is an interesting direction for future work. But our current results already show that a fixed $\alpha$ is sufficient to consistently outperform all baselines in Table 2. PL degrades smoothly as contamination rates increase (Figures 4–6, Appendix D). Thus, PL is reasonably robust to the exact choice of alpha even without an elaborate scheduler.
>
> **[1]** Elenter, Juan, Navid NaderiAlizadeh, and Alejandro Ribeiro.
> "A Lagrangian duality approach to active learning." NeurIPS, 2022.
>
> **[2]** Loshchilov, Ilya and Frank Hutter.
> "SGDR: Stochastic gradient descent with warm restarts."
> arXiv preprint arXiv:1608.03983, 2016.
>
> > **Question 4: Generalization to Unknown Attacks**
> >
> > - Does PL remain effective against adaptive or unknown trigger attacks that were not seen during fine-tuning?
> > - How does it perform on semantic or dynamic triggers instead of static patterns?
>
> To assess cross-attack generalization, we use the following protocol: (1) start with a model trained under attack type A; (2) fine-tune it using PL ($\alpha = 0.2$) with samples attributed to attack type B, and (3) test the resulting model against the original attack type A. We repeat this for all pairs of distinct attack types under four types of contamination cases: perfect partition (no contamination), benign contamination (FN only), malicious contamination (FP only), and combined contamination (FN+FP).
>
> The four $5\times5$ cross-attack tables (Tables 4-5, Appendix E) show: under perfect partition and benign contamination, clean accuracy remains high and stable across all unseen attacks. ASR remains low when PL is not trained on the target attack, often matching or beating the model fine-tuned directly on that attack. PL generalizes to unseen backdoor triggers in such scenarios. Under malicious and combined contamination, some degradation occurs, but there are no catastrophic failures.
>
> | attack   | blended     | blended_adaptive | ssba         | visible      | wanet       |
> |----------|-------------|------------------|--------------|--------------|-------------|
> | blended  | 0.795/0.006 | 0.795/0.006      | 0.795/0.007  | 0.795/0.009  | 0.795/0.010 |
> | blended_adaptive | 0.785/0.012 | 0.785/0.012 | 0.785/0.007 | 0.785/0.008 | 0.785/0.010 |
> | ssba     | 0.774/0.007 | 0.774/0.007      | 0.774/0.013  | 0.774/0.008  | 0.774/0.007 |
> | visible  | 0.774/0.001 | 0.774/0.001      | 0.774/0.002  | 0.774/0.021  | 0.774/0.002 |
> | wanet    | 0.771/0.001 | 0.771/0.001      | 0.771/0.003  | 0.771/0.005  | 0.771/0.008 |
>
> PL is not restricted to a static trigger. Finetuning on WaNeT and SSBA demonstrates that PL generalizes to dynamic, input-dependent transformation (Tables 2–3, main §4.2).  Evaluation of the semantic trigger is promising for future work. Existing results indicate PL is effective across diverse trigger types, including foundational visible patch triggers, adaptive blended triggers, spatially warped triggers (WaNet), and blended content triggers (SSBA).

---

> ### Author Response · Authors · 2025-11-20
>
> > **Question 5: Comparison with Modern Defenses**
> >
> >- How does PL perform compared to exposure-based, representation-level unlearning, or causal filtering defenses introduced in 2024–2025?
> >- Can PL be combined with these methods to form a hybrid, more resilient defense?
>
> Recent work in 2024-2025 introduced backdoor defense that focuses on exposure, representation-level unlearning, and causal filtering. For example,  EBYD [1] proposes an exposure-based framework that first exposes triggers via model preprocessing and then applies a detection and removal method to identify and eliminate the backdoor features. Causality-inspired Backdoor Defense (CBD) [2] uses structural causal models to disentangle confounding trigger–label associations and learn deconfounded feature representations. These model targets different components of the model pipeline.
>
> In contrast, PL is a fine-tuning objective that requires a coarse partition between possibly benign and possibly malicious samples. PL is orthogonal to the above categories; it can be readily combined with them. In our experiments, PL also works seamlessly with existing defenses such as feature-shift tuning (FST), pruning, and the detection splits method. Thus, PL could also serve as a drop-in module to enhance performance rather than a replacement.
>
> [1] Li, Yige et al. “Expose Before You Defend: Unifying and Enhancing Backdoor
>     Defenses via Exposed Models.” arXiv:2410.19427, 2024.
>
> [2] Zhang, Zaixi et al. “Backdoor Defense via Deconfounded Representation Learning.”
>      CVPR, 2023.

---

### Official Review · Reviewer_XN8y · 2025-10-31

**Soundness:** 3
**Presentation:** 2
**Contribution:** 3
**Rating:** 4
**Confidence:** 3

**Summary:**

This paper proposes a post-training backdoor defense method that leverages both benign and maliciously labeled samples. Instead of discarding suspicious data, the method formulates distinct loss objectives for benign and malicious samples, introducing a “forgetting signal” to remove trigger-related representations. The approach is theoretically grounded and experimentally validated on several benchmark datasets.

**Strengths:**

1.The idea of actively forgetting triggers via loss optimization, guided by labeled malicious samples, is original and contrasts well with existing “sample rejection” paradigms.

2.The paper provides formal proofs that justify the convergence and effectiveness of the proposed mechanism.

3.Clear motivation: The authors identify a real-world issue—data contamination in fine-tuning—and propose a method that attempts to address it in a principled way.

**Weaknesses:**

1.Semantic inconsistency: The paper first claims that a clean fine-tuning set is unrealistic due to possible contamination, but later assumes access to “clean samples.” The notion of “clean data” thus requires clarification—perhaps as “a mostly clean but partially contaminated sample pool.”

2.Limited deployment realism: The proposed defense assumes post-hoc identification of malicious inputs through monitoring or human response. This makes it a reactive mechanism, which may not prevent immediate harm in real-world attacks.

3.Narrow experimental scope: Experiments cover only single-target, explicit-trigger backdoor settings. The method’s performance under multi-target, multi-trigger, or stealthy (implicit) attacks remains unexplored.

**Questions:**

1.How robust is the proposed method when the “clean” pool is partially contaminated?

2.Could the defense mechanism be adapted to an online or proactive setting, rather than relying on post-attack detection?

---

> ### Author Response · Authors · 2025-11-20
>
> We appreciate the reviewer's thoughtful evaluation and constructive suggestions, which guided us to strengthen several technical explanations in the revision!
>
> > Weakness 1: Semantic inconsistency: The paper first claims that a clean fine-tuning set is unrealistic due to possible contamination, but later assumes access to “clean samples.” The notion of “clean data” thus requires clarification—perhaps as “a mostly clean but partially contaminated sample pool.”
>
> Thank you for pointing out this inconsistency. We have taken the time to revise the terminology we use in the paper to ensure clearer and more consistent use of “clean” versus “mostly-clean” data.
>
> In practice, it is unrealistic to obtain a perfectly clean dataset. The supposedly clean fine-tuning set in the real world may contain a small portion of poisoned samples even after preprocessing. Existing state-of-the-art backdoor detection methods [1, 2, 3, 4, 5, 6] can significantly reduce but not eliminate poisoned samples, achieving sensitivity and specificity above $0.5$. These detectors can be used before fine-tuning to construct a mostly-clean and mostly-poisoned partition. Our method operates on these imperfect partitions and only assumes that the “mostly-clean” partition contains fewer than half malicious samples, and the “mostly-poisoned” partition contains more than half malicious samples. This mild assumption aligns with any detector whose sensitivity is $>0.5$.
>
> **[1]** Dong, Yinpeng et al. “Black-box detection of backdoor attacks with limited information and data.” ICCV 2021.
> **[2]** Xian, Xun et al. “A unified detection framework for inference-stage backdoor defenses.” NeurIPS 2023.
> **[3]** Xu, Xiaoyun et al. “BAN: detecting backdoors activated by adversarial neuron noise.” NeurIPS 2024.
> **[4]** Hu, Mengxuan et al. “BBCaL: Black-box Backdoor Detection under the Causality Lens.” TMLR 2024.
> **[5]** Li, Wei et al. “PSBD: Prediction shift uncertainty unlocks backdoor detection.” CVPR 2025.
> **[6]** Guan, Zihan et al. “UFID: A Unified Framework for Black-box Input-level Backdoor Detection.” AAAI 2025.
>
>
> > Weakness 2: Limited deployment realism: The proposed defense assumes post-hoc identification of malicious inputs through monitoring or human response. This makes it a reactive mechanism, which may not prevent immediate harm in real-world attacks.
>
> Backdoor defenses differ significantly depending on the threat model, and our work targets the model-level threat setting. It reflects a common real-world situation where practitioners fine-tune publicly released or third-party pretrained models (e.g., via HuggingFace, GitHub, or industrial model hubs). In this case, post-hoc correction is the only feasible class of defenses, because the backdoor is already embedded in the model before fine-tuning begins.
>
> Our method does not assume a human operator who must identify malicious inputs. Instead, it is compatible with existing automated state-of-the-art detection techniques discussed above, which can automatically surface suspicious samples. These detectors can be combined with our fine-tuning procedure to remove backdoor functionality before deployment, rather than reacting only after harm occurs. Meanwhile, this deployment assumption applies broadly to all post-hoc model-level backdoor defenses, including many baselines we cite (FST [1], FT [1], SAM [2], IBAU [3]).
>
> **[1]** Min, Rui et al. Towards Stable Backdoor Purification through Feature Shift Tuning (FST). NeurIPS, 2023.
> **[2]** Foret, Pablo et al. Sharpness-Aware Minimization for Efficiently Improving Generalization. ICLR, 2021.
> **[3]** Huang, Zeyu et al. I-BAU: Indirect Backdoor Adversarial Unlearning. ICLR, 2022.
>
>
> > Weakness 3: Narrow experimental scope: Experiments cover only single-target, explicit-trigger backdoor settings. The method’s performance under multi-target, multi-trigger, or stealthy (implicit) attacks remains unexplored.
>
> We have shown that our proposal outperforms the current state-of-the-art finetuning backdoor defense against some single-target attacks, including a stealthy (invisible trigger) attack (WaNet). Evidence appears in Table 1 (main text §4.1), Table 2 (main text §4.2), and additional experiments in Appendix C. Although we leave multi-target and multi-trigger attacks to future work, we assume access only to samples that have been partitioned so that fewer than half of the malicious samples are in the mostly-clean partition. Hence, we make no assumptions about the trigger or target data amounts, and our proposal can be directly applied to these settings.

---

> ### Author Response · Authors · 2025-11-20
>
> > Question 1: How robust is the proposed method when the “clean” pool is partially contaminated?
>
> Our method is explicitly designed for scenarios where the perfectly clean or poisoned pools are infeasible. Thus, we have mostly clean and mostly-poisoned fine-tuning sets. Our experiments include benign contamination: FN, malicious contamination: FP, and both types of contamination. We characterize how performance changes when moving from perfectly partitioned data to partially contaminated partitions in Tables 2–3 (main §4.2) and Figures 4–9 (Appendix D). Across all contamination settings, PL consistently demonstrates performance dominance over the other baselines.
>
>
> > Question 2: Could the defense mechanism be adapted to an online or proactive setting, rather than relying on post-attack detection?
>
> Although outside the scope of our paper, you provide an interesting avenue for future research. One of our previous responses addressed the proactive setting, noting that our method could be applied to either the original training dataset (or a validation/testing split) or a synthetic dataset. Furthermore, for the online setting, we suggest that after each observation, we could perform (stochastic) gradient descent (or accent) on a single observation at a time (online) based on any detection model that could classify the sample as “clean” or “malicious”. Another possible (related) extension could be soft partitioning (using the probability that a sample is “malicious”) to weight the gradient rather than determining descent vs. weighted ($\alpha$) ascent.

---

### Official Review · Reviewer_Xpmc · 2025-11-01

**Soundness:** 3
**Presentation:** 4
**Contribution:** 4
**Rating:** 6
**Confidence:** 3

**Summary:**

While existing backdoor defense finetuning assumes access to an absolutely clean set, this is not realistic in practice. Thus, the authors propose an algorithm to take potentially contaminated data as inputs to achieve promising performance.
The authors first design a partition loss that takes potentially contaminated benign and malicious datasets, and correspondingly designed an algorithm partition loss finetuning that optimizes the partition loss. PLFT is proved to be effective through comprehensive evaluations.

**Strengths:**

(1) The problem is well motivated.

(2) The partition loss is elegantly defined, and the partition loss finetuning algorithm is designed to effectively lowers the loss. It is also well proved through theory and supported by experiment results.

(3) The evaluation  is comprehensive.

**Weaknesses:**

My major concern on this paper is the correlation between \alpha and contamination rates in both benign and malicious set - intuitively I believe there's a correlation among them. Specifically, I think the paper is more complete if:

(1) The correlation of the parameters is proved theoretically (you can probably add a subsection in the method section)

(2) The correlation is studied in the evaluation.

**Questions:**

See Weakness section.

---

> ### Author Response · Authors · 2025-11-20
>
> We thank the reviewer for insightful comments that helped us clarify key aspects of our methodology!
>
> In our method, $\alpha$ serves as a regularization weight that controls the penalty applied to malicious behavior in $D_m$​. Its role is not determined directly by the contamination rate but by the balance between preserving benign behavior and suppressing malicious behavior. As the contamination rate increases:
> 1. The proportion of poisoned data in $D_m$ decreases, so we would want a larger $\alpha$ to work harder to remove the poisoned behavior.
> 2. The proportion of benign data in $D_m$ increases, so we would want a smaller $\alpha$ to prevent degradation to benign performance.
>
> For this reason, we believe there may not necessarily be a relationship between $\alpha$ and the contamination rate.
> As experimental evidence, the value of $\alpha = 0.2$ at which we outperformed competing methods was the same across different contamination levels.
> These results are reported in Appendix D (Figures 4–9), where PL remains robust even as contamination varies from 5% to 20%.

---

### Author Response · Authors · 2025-12-02
**Summary of Reviewer Consensus and Our Response for Submission 18450**

Dear Area Chair,

Thank you for handling our submission. We write to summarize the discussion-phase outcome: although not all reviewers replied, **the reviewer who initially raised the most significant concerns replied that our rebuttal successfully addressed their primary issues and that the scientific contribution and presentation have been substantially improved**.

We propose **Partition-Loss Fine-Tuning (PL)**, a simple and practical post-training backdoor defense that leverages mostly clean and flagged malicious sets. By jointly minimizing the benign loss and maximizing the malicious loss, PL explicitly unlearns the trigger–target association and remains robust even when the fine-tuning data is partially contaminated.

## **Reviewer Consensus on Strengths**

Across all four reviewers, there is clear agreement on the novelty and practicality of our work:
- Problem Importance & Practicality: Reviewers **Xpmc** and **XN8y** agreed that the paper addresses a well-motivated and realistic problem. Reviewer **UZFP** further highlighted PL’s simplicity: it requires no pre-training data; it is easy to implement, and remains effective under contaminated fine-tuning.
- Novelty: Reviewer **XN8y** found the optimization-based forgetting mechanism to be original and distinct from existing sample-rejection paradigms.
- Strong Empirical Validation: Reviewer Xpmc praised the comprehensive evaluation. Reviewer **UZFP** emphasized that PL consistently achieves low ASR with strong C-ACC across settings.
- Theoretical Soundness: Reviewers Xpmc and **XN8y** both highlighted the clarity and usefulness of the theoretical analysis supporting the PL objective.


## **Summary of Response Content**

**Reviewer Xpmc: Parameter Sensitivity**

-  **Concern: Relationship between \alpha and contamination rates**. We clarified that \alpha balances clean accuracy and backdoor forgetting. But it is not tied to the contamination rate. New contamination-sweep experiments (Appendix D, Figs. 4–9) show that PL remains stable from 5%–20% contamination. \alpha = 0.2 consistently performs well across all settings.

**Reviewer XN8y: Robustness & Deployment Practicality**
-  **Concern 1: Robustness under contaminated “clean” pool**. PL is designed for imperfect partitions. Experiments with benign (FN), malicious (FP), and mixed (FN+FP) contamination (Tables 2–3,§4.2; Figures 4–9, Appendix D), show that PL consistently outperforms baseline defenses from 5% to 20% contamination.

- **Concern 2: Deployment realism**. We clarified that PL matches common practice: practitioners fine-tune publicly released or third-party pretrained models. They must remove embedded backdoors before deployment. PL can naturally integrate with automated detectors, avoiding reliance on human monitoring.

**Reviewer UZFP: Assumption & Generalization**
- **Concern 1: Dependence on trigger knowledge and perfect partitioning**. PL does not require knowing the trigger or target class. It only assumes a detection model with accuracy above random chance. Extensive experiments confirm low ASR and strong C-ACC even under imperfect partitions.

- **Concern 2: Generalization and modern defenses.** We added cross-attack experiments (Appendix E, Tables 4–5). PL maintains high C-ACC and low ASR against unseen triggers, including dynamic attacks. We also clarified that PL is orthogonal to exposure-based, representation-level, and causal defenses, and can be combined with them.

**Reviewer ZcSL: Threat Model, Contamination Levels, and Additional Details**
- **Concern 1: Realism of the threat model**. We clarified that PL operates in the incident-response phase, where modern backdoor detection methods (e.g., NC, B3DSS, BAN, BBCaL, PSBD, UFID) achieve ≥95% recall. Having access to a small, noisy, malicious pool is hence realistic.

- **Concern 2: Contamination rates**. We added contamination-sweep experiments from 5%–20%. Across FP, FN, and FP+FN settings, PL degrades smoothly; moderate \alpha (0.1–0.5) remain effective.

## **Summary of Changes to the Submitted Paper**
1. **Clarified terminology** (“clean” → “mostly clean”): we updated the abstract and introduction to reflect that our benign pool allows contamination.
2. **Added references** supporting access to malicious samples: we included citations to a recent high-accuracy backdoor detector in the introduction.
3. **Added contamination-sweep experiments**: we added new results at 5%, 10%, 15%, and 20% contamination across FP, FN, and FP+FN settings in Appendix D, Figs. 4–9.
4. **Added cross-attack generalization experiments**: we added a full set of 5×5 cross-attack evaluations in Appendix E, Tables 4–5. Results show that PL maintains strong performance on unseen trigger types, including dynamic triggers.

We hope this summary assists in your final decision-making process.

Best regards,

Authors of #18450

---

### Meta-Review · Area_Chair_jRXB · 2026-01-09

**Summary:**

This paper proposes Partition-Losses Fine-Tuning (PL), a post-training backdoor based on finetuning. Instead of assuming a completely clean finetuning data, PL can operate on both benign and flagged malicious samples with an optimization objective that minimizes the loss on clean data and maximizes the loss on malicious samples, effectively unlearning the trigger-to-target association. Extensive experiments across multiple datasets and attacks show PL’s effectiveness, even when the fine-tuning data is partially contaminated.

The reviewers are generally concerned about the sensitivity of the method’s performance on the potentially contaminated finetuning data (specifically when the detector fails), and the lack of evaluation on SOTA baselines. Other concerns, such as whether the defense generalizes to various trigger types or unknown attacks and theoretical discussion, have been clarified. Nevertheless, I believe that the paper’s empirical analysis is still limited, e.g., single-target attacks beyond WaNet, such as latent-preservation or several clean-label attacks, or multi-target and multi-trigger attacks are not included. This makes it very difficult to comprehensively understand the behavior of the methods against possible types of attacks. The paper, therefore, is not ready yet for publication and I hope the authors will revise the paper accordingly based on these comments.

**Reviewer Concerns:**

- Xpmc:
   - correlation between \alpha and contamination rates in both benign and malicious set —> should be theoretically proved or studied in evaluation
- XN8y
   - The paper first claims that a clean fine-tuning set is unrealistic due to possible contamination, but later assumes access to “clean samples”
   - The proposed defense assumes post-hoc identification of malicious inputs through monitoring or human response
   - Experiments cover only single-target, explicit-trigger backdoor settings.
- UZFP
   - dependence on prior knowledge of the backdoor trigger and target label, as well as its vulnerability to data partitioning errors.
   - PL lacks dynamic control and trigger-agnostic robustness.
   - parameter sensitivity
   - generalization to unknown attacks
   - comparisons with modern defenses
- ZcSL
   - unrealistic threat model of clean sample assumption
   - lack of theoretical discussion on $D_m$ and $D_b$ and upperbound/lowerbound validation
   - lack of SOTA approaches

Most concerns have been addressed, except for the lack of evaluation on several other types of attacks and comparisons against defenses.

**Reviewer Scores:**

Reviewer ZcSL seems to promise improving the score, but some concerns have not sufficiently addressed. For other reviewers, some concerns have not been addressed, so they are unlikely to increase their ratings.
    - Xpmc: 6
    - XN8y: 4
    - UZFP: 4
    - ZcSL: 2 — potentially improve score

---

### Decision · Program_Chairs · 2026-01-26

Reject